# Neural correlates of perceptual similarity masking in primate V1

Spencer Chin-Yu Chen[1,2,3,4,5], Yuzhi Chen[1,2,3,4], Wilson S Geisler[1,2,3], Eyal Seidemann[1,2,3,4]*

[1]Center for Perceptual Systems, University of Texas at Austin, Austin, United States; [2]Department of Psychology, University of Texas at Austin, Austin, United States; [3]Center for Theoretical and Computational Neuroscience, Austin, United States; [4]Department of Neuroscience, University of Texas at Austin, Austin, United States; [5]Department of Neurosurgery, Rutgers University, New Brunswick, United States

*For correspondence: eyal@austin.utexas.edu

**Competing interest:** The authors declare that no competing interests exist.

**Sent for Review** 14 June 2023
**Preprint posted** 10 July 2023
**Reviewed preprint posted** 18 September 2023
**Reviewed preprint revised** 13 March 2024
**Version of Record published** 09 April 2024

**Abstract** Visual detection is a fundamental natural task. Detection becomes more challenging as the similarity between the target and the background in which it is embedded increases, a phenomenon termed 'similarity masking'. To test the hypothesis that V1 contributes to similarity masking, we used voltage sensitive dye imaging (VSDI) to measure V1 population responses while macaque monkeys performed a detection task under varying levels of target-background similarity. Paradoxically, we find that during an initial transient phase, V1 responses to the target are enhanced, rather than suppressed, by target-background similarity. This effect reverses in the second phase of the response, so that in this phase V1 signals are positively correlated with the behavioral effect of similarity. Finally, we show that a simple model with delayed divisive normalization can qualitatively account for our findings. Overall, our results support the hypothesis that a nonlinear gain control mechanism in V1 contributes to perceptual similarity masking.

## eLife assessment

This **important** study used Voltage Sensitive Dye Imaging (VSDI) to measure neural activity in the primary visual cortex of monkeys trained to detect an oriented grating target that was presented either alone or against an oriented mask. The authors show **convincingly** that the initial effect of the mask ran counter to the behavioral effects of the mask, a pattern that reversed in the latter phase of the response. They interpret these results in terms of influences from the receptive field center, and although an alternative view that emphasizes the role of the receptive field surround also seems reasonable, this study stands as an interesting contribution to our understanding of mechanisms of visual perception.

## Introduction

Searching for, and detecting, visual targets in our environment is a ubiquitous natural task that our visual system performs exceptionally well. A key feature of behavioral detection performance is that the texture similarity between the target and the background in which it is embedded profoundly affects target detectability. The more similar are the target and the background, the harder it is to detect the target (*Campbell and Kulikowski, 1966*; *Foley, 1994*; *Sebastian et al., 2017*; *Stromeyer and Julesz, 1972*; *Watson and Solomon, 1997*; *Wilson et al., 1983*). This phenomenon, which is termed 'similarity masking', is the foundation of camouflage.

An example of similarity masking is illustrated in *Figure 1*. Detecting a low contrast oriented visual target is easy on a uniform gray background (*Figure 1A*). Detectability decreases when the target has

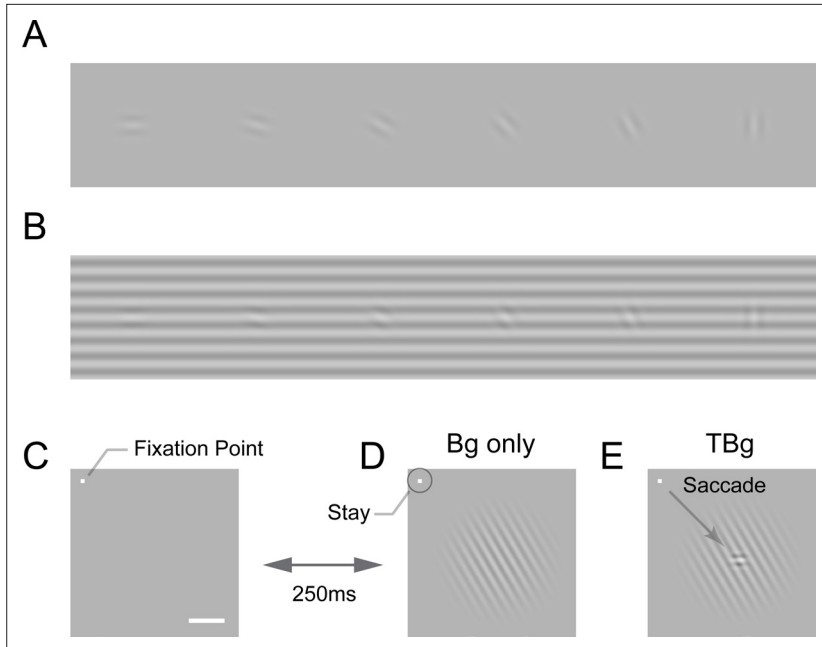

**Figure 1.** Target-background similarity masking and behavioral task. (**A**) Low contrast orientated targets can be easily detected on uniform background. (**B**) Similarity masking is induced by an orientated background. The same additive target from (**A**) becomes hard to detect when target orientation matches background orientation (see also perceptual demonstration in *Supplementary file 1*). (**C–E**) Orientation masking was assessed in two awake behaving macaque monkeys performing a target detection task. Monkey commence the task by fixating at the small bright square (**C**). A few moments later, a 4° raise-cosine-masked background grating was flashed at ~3° eccentricity for target detection (**D**). The horizontal white bar represents one degree of visual angle. In 50% of the trials, a small additive horizontal Gabor target was also added to the background (**E**). The monkey indicated the presence of the target by making a saccade to the target location, and indicated target absent by maintaining gaze at the fixation point. The Gabor target was always the same – a cosine centered, horizontal Gabor at 4cpd on 0.33° FWHM envelope. The background grating was also cosine-centered at 4cpd such that the background completely aligned with the target when they were the same orientation (as in **B**). Orientation of the grating ranged from 0° to 90° with respect to the Gabor target and was randomized between trials. Bg – background; TBg – target plus background.

a similar orientation as the background (*Figure 1B*). The neural basis of similarity masking is not well understood. The main goal of the current study was to test the hypothesis that neural interactions between the representations of the target and background in the primary visual cortex (V1) contribute to the perceptual effect of similarity masking.

The responses of visual neurons to a target can be strongly modulated by the context in which the stimulus is presented. Such contextual modulations have powerful, complex and diverse effects in the visual cortex (*Allman et al., 1985*; *Angelucci et al., 2017*; *Angelucci and Bressloff, 2006*; *Bai et al., 2021*; *Cavanaugh et al., 2002b*; *Henry et al., 2020*; *Michel et al., 2018*; *Sceniak et al., 1999*; *Shushruth et al., 2012*). Most of these effects reflect sublinear interactions between the target and the background, suggesting that they could potentially contribute to behavioral masking effects. If nonlinear computations in V1 contribute to similarity masking, we would predict that the signals evoked by a target will be maximally reduced by a background that is similar to the target.

We tested this hypothesis by measuring V1 population responses in macaque monkeys while they performed a visual detection task under masking conditions (*Figure 1C–E*). Because the nature of contextual modulations in V1 is complex, a second goal of our study was to quantitatively characterize the spatiotemporal dynamics of V1 population responses to different combinations of targets and backgrounds.

As a first step, we characterized the behavioral effects of similarity masking in two macaque monkeys, demonstrating clear effects of similarity on target detectability and reaction times. These results confirm that macaque monkeys are a good animal model for human similarity masking.

Second, we used voltage-sensitive dye imaging (VSDI; *Grinvald and Hildesheim, 2004*; *Seidemann et al., 2002*; *Shoham et al., 1999*) to measure V1 population responses at two scales: the scale of the retinotopic map and the scale of orientation columns, while the monkeys performed the similarity masking detection task. To study the effect of similarity masking on the neural detection sensitivity, we constructed a task-specific decoder at each scale. Each decoder first pools the responses using a scale-dependent spatial template and then combines these responses over time to form a decision variable. The distributions of the decision variable in target-present vs. target-absent trials are used to compute neural sensitivity that can be compared to behavioral sensitivity (*Seidemann and Geisler, 2018*).

We found that V1 population responses to the target and background display two distinct phases. An initial transient phase that starts at response onset, and a second phase that lasts until stimulus offset or the animal's response. Surprisingly, the first phase displays a paradoxical effect; during this phase the target evoked response is strongest when the target and background are similar and is therefore anti-correlated with behavior. This effect reverses in the second phase so that in this phase the target-evoked response is reduced with increased target-background similarity. V1 responses during this second phase are therefore consistent with behavior.

We also observed complex spatiotemporal dynamics of the population response to the target and background stimuli, including a repulsion of V1 columnar-scale representation of target orientation in the direction away from the background orientation.

Finally, we show that a simple dynamic population gain control model can qualitatively account for our physiological and behavioral results, and that the estimated properties of the gain-control mechanism are consistent with a principled computational approach to feature encoding and decoding. Overall, our results are consistent with the hypothesis that contextual interactions between the representations of the target and background in V1 are likely to contribute to the perceptual phenomena of similarity masking.

## Results

### Behavioral effect of target-background similarity masking

To study the neural basis of visual similarity masking, we trained two monkeys (*Macaca mulatta*) to perform a visual detection task in which a small horizontal target appeared on a larger background at a known location in half of the trials (*Figure 1E*). The monkey indicated target absence by maintaining fixation and target presence by making a saccadic eye movement to the target location as soon as it was detected. Within a block of trials, the contrast of the target and the background were fixed, while the orientation of the background varied randomly from trial to trial, allowing us to test for the effect of target-background orientation similarity on behavioral and neural detection sensitivities.

We tested the behavioral effect of similarity masking over five combinations of target and background contrasts (*Figure 2A*). For each combination, we measured the behavioral sensitivity as a function of background orientation. Performance with no background (uniform gray screen) served as a baseline (*Figure 2A*, dashed horizontal lines). Performance as a function of background orientation was fitted with an inverted Gaussian function. At all five target and background contrast combinations, detection sensitivity was lowest when background and target orientations matched, confirming the expected similarity masking effect from human subjects. These results demonstrate that macaque monkeys are a good animal model for studying the neural basis of human similarity masking.

The supplementary information includes a perceptual demonstration of similarity masking for a wide range of target amplitudes, orientations, and spatial-frequencies (*Supplementary file 1*). This demonstration can give the reader an intuitive sense of the masking effects studied here.

We also examined the effect of target-background orientation similarity on the monkeys' reaction times (*Figure 2C*). We find two distinct effects of orientation similarity on reaction times. At higher target and background contrasts, reaction times are maximal when the background and target have the same orientation (when detectability is lowest and the task is hardest) and monotonically decrease as target-background similarity decreases (detectability increases and the task becomes easier). Surprisingly, at lower target and background contrasts, reaction times are low when the background matches target orientation, then increases as the background-target orientation difference increases, and then drops again when the background approaches the orthogonal orientation to the target. Thus, under these

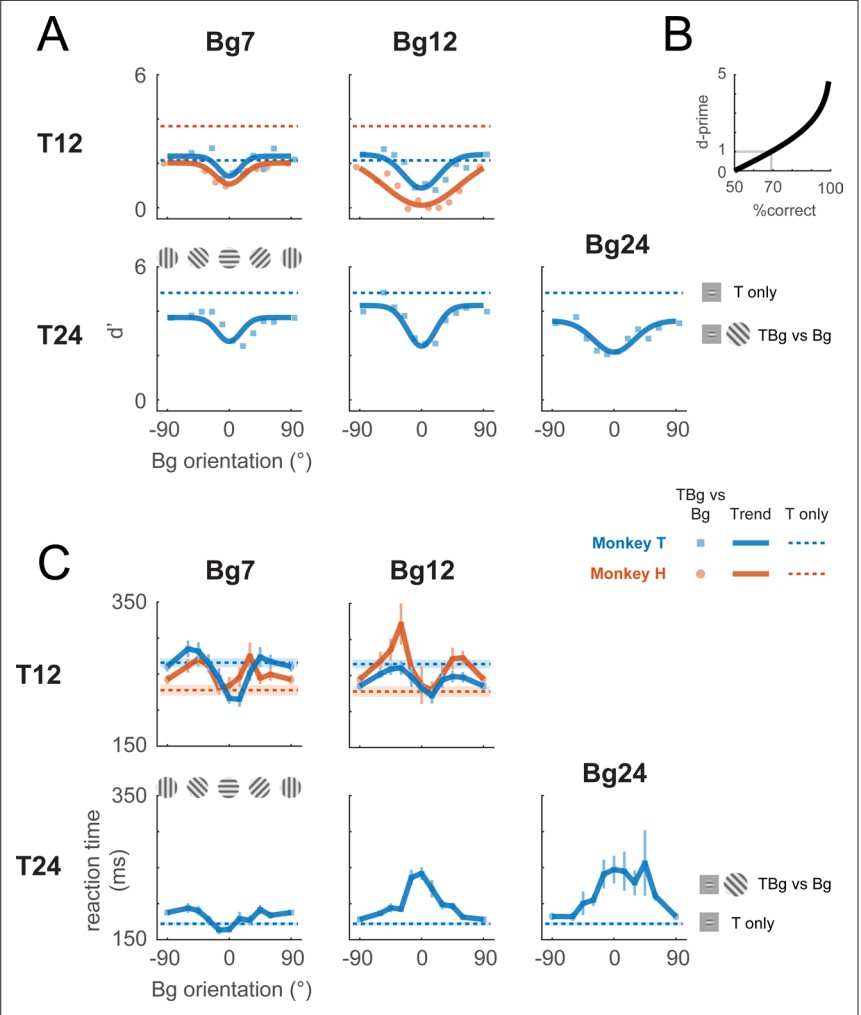

**Figure 2.** Behavioral effect of target-background similarity masking. (**A**) Target detection performance in monkeys was affected by the orientation of the background grating over a variety of target contrast (T## = ##% contrast target) and background contrast levels (Bg## = ##% contrast background). Signal detection measure d-prime (d′) of the target is plotted for uniform background (dotted lines), and for each background orientation (markers). In most cases, there was a general performance reduction from uniform background to a grating background. Additionally, performance was further reduced when the background orientation was more aligned to the target (0°). A fitted Gaussian (solid line) illustrates the performance change due to orientation masking. d′ was calculated from the hit rate (correctly reporting target present) and the false alarm rate (reporting target present when it was absent). The relationship between d′ and optimum performance level in percent correct is plotted in (**B**). (**C**) Reaction time – calculated from stimulus onset to saccade initiation for Hit trials – is plotted for uniform background (dotted lines) and for each background orientation (solid lines). Error bars indicate the standard error of the mean. Data were pooled within each monkey across experiments. Each experiment contains a single combination of target and background contrast levels, with uniform background and orientated background trials assessed in separate blocks.

conditions, we see an interesting decoupling between difficulty and reaction time, so that reaction times can be shortest in the harder conditions. This surprising effect is present in both monkeys. Some of the complex neural dynamics described below could explain this interesting effect (see Discussion).

Our next goal was to test the hypothesis that contextual interactions between the representations of the target and background in V1 contribute to the observed behavioral similarity masking results.

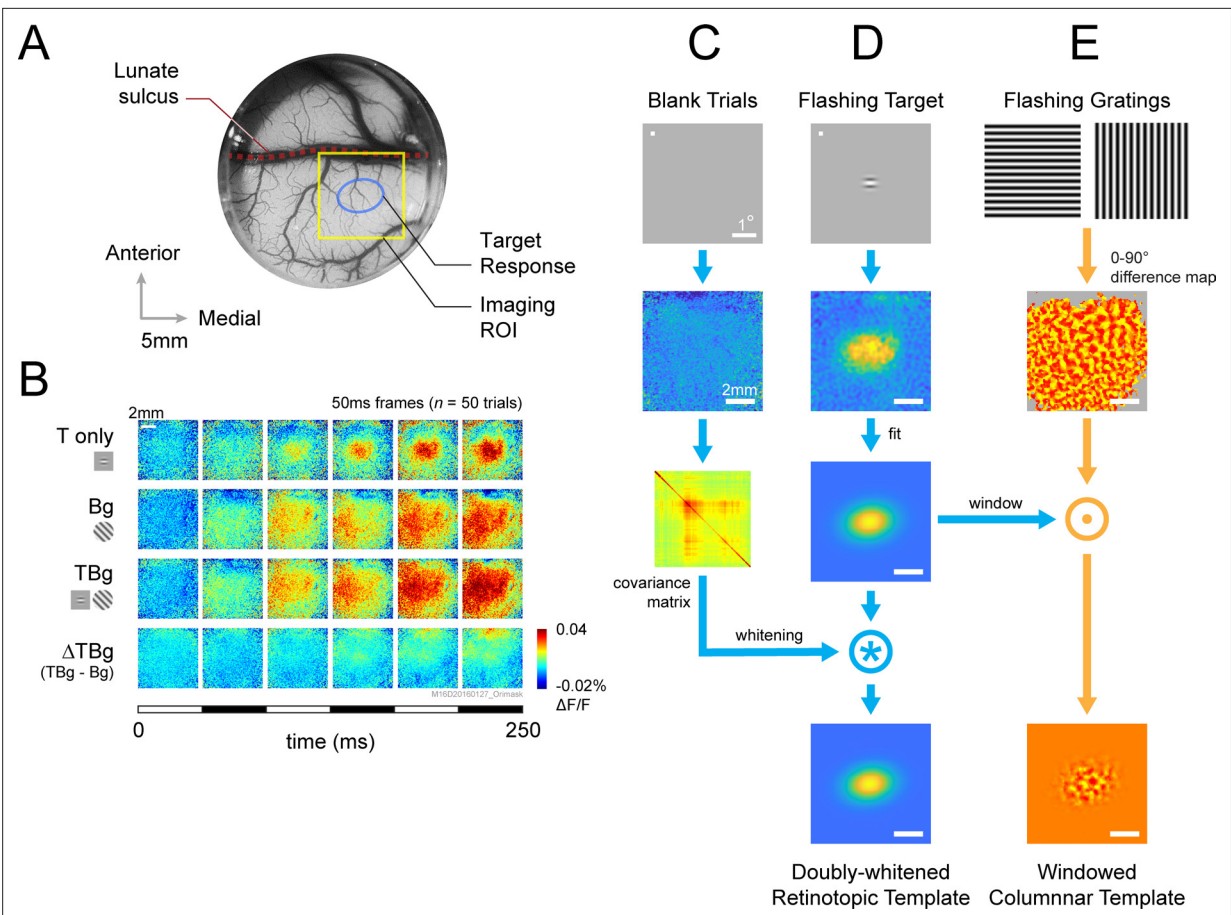

**Figure 3.** VSD response and decoder schematic. (**A**) A cranial chamber and a transparent artificial dura provided chronic imaging access to V1 of the monkey. V2 is completely hidden in the lunate sulcus based on the retinotopic map taken in a separate imaging session (*Figure 3—figure supplement 1*). Imaging ROI and target response at 2 SD of the fitted Gaussian from the example recording in **B-E** are illustrated. (**B**) Example recording of the voltage-sensitive dye (VSD) response in an 8x8 mm imaging region-of-interest (ROI). Response to the Gabor target on uniform background could be easily identified in VSD (top row). The visual span of the background grating extended beyond the coverage of the imaging window, evoking an encompassing response over the entire ROI (2nd row). When the background was presented with the additive target, the response to the target was diminished (3rd and 4th rows). Each VSD response map were averaged across 50 trials, over 5 frames captured at 100 Hz. T only – target only; Bg – background only; TBg – target and background; ΔTBg – target and background minus background only. (**C**) Target response was extracted at the retinotopic scale by estimating its response profile with a two-dimensional Gaussian. The profile was estimated from response from a separate recording block on each experiment day. To optimized signal-to-noise, in this recording block, the target was flashed repeatedly at 5 Hz while the monkey maintained fixation. The effect of spatially correlated VSD noise was minimized by estimating a whitening kernel from trials without stimulus presentation (see Methods). (**D**) Target response was extracted at the columnar scale by estimating the orientation map within the imaging area. This was constructed from full-field gratings flashed at 5 Hz in a separate recording block on each experiment day (see Methods). The columnar map in the 0°–90° axis was extracted and windowed down to the retinotopic profile to identify the columnar scale response of the target.

The online version of this article includes the following figure supplement(s) for figure 3:

**Figure supplement 1.** Retinotopy of imaging chambers and target placement positions.

## Neural population responses to target and background stimuli in macaque V1

While the monkeys performed the similarity masking detection task, we used VSDI to measure V1 population responses to the target and the background. In each cranial window, we first used a fast and efficient VSDI protocol to obtain a detailed retinotopic map (*Yang et al., 2007*). We then positioned the target so that its neural representation fell at the center of our imaging area.

The target elicits V1 population activity at two fundamental spatial scales. At the large retinotopic scale, the target evokes an activity envelope that spreads over several mm² and is well fitted by a two-dimensional (2D) Gaussian (*Figure 3B*, top row; *Chen et al., 2006*; *Chen et al., 2012*; *Sit et al.,*

*2009*). Our 8x8 mm$^2$ imaging area allows us to capture this entire target-responsive region. Because the background is much larger than the target and is centered at the same location in the visual field, it produces a relatively uniform response within the imaging area (*Figure 3B*, second row). Similarly, the target-plus-background stimulus elicits activity within the entire imaged area, with a relatively elevated activity at the retinotopic region corresponding to the target location (*Figure 3B*, 3rd row). However, the target-evoked response in the presence of the background (response to target plus background minus response to the background alone) appears significantly weaker than the response to the target alone (*Figure 3B*, 1st vs. 4th row). This reduced target-evoked response in the presence of the background could contribute to the perceptual masking effect of the background. Our goal here was to determine how this sublinear interaction between the response to the target and background depends on target-background similarity in orientation.

In addition to the retinotopic-scale response envelope, fine scale response modulations at a scale of individual orientation columns (width of ~0.3 mm) reflect the orientation of the target and background. These columnar scale modulations have a relatively small amplitude and therefore appear as small ripples riding on top of the larger retinotopic response envelope. The relatively smaller VSDI responses at the columnar scale is due to a mixture of robust non-orientation selective V1 population responses in V1 as well as optical and biological blurring (*Chen et al., 2012*). We can selectively access the columnar scale signals by spatially filtering the responses at the scale of the orientation columns. Despite their small relative amplitude, these columnar-scale signals provide high-quality single-trial orientation decoding (*Benvenuti et al., 2018*).

## Retinotopic-scale effect of target-background similarity masking

To study the effect of similarity masking on V1 responses at the retinotopic scale, we used an optimal linear decoder of V1 population responses (*Chen et al., 2006*; *Chen et al., 2008*) that allows us to assess the neural detection sensitivity of V1 population responses (i.e. how well one can detect the target from single-trial V1 population responses) (*Figure 3C–D*). The retinotopic decoder takes into account the location and shape of the envelope of the target-evoked response (*Figure 3D*), as well as the structure of the noise covariance matrix (*Figure 3C*; *Chen et al., 2006*).

*Figure 4* summarizes the dynamics of the retinotopic template output in response to the V1 signals across all of our experiments for two combinations of target and background contrasts (see *Table 1* and *Figure 4—figure supplement 1* for the full set of tested target/background combinations). When presented on a uniform gray background, the target-related retinotopic signal begins to rise ~40 ms after target onset, reaches its peak ~100 ms after stimulus onset, and remains high for the next 100ms (*Figure 4B and H*, black curve). However, when the same target is added to the background, the target-related retinotopic signals display a wide range of responses that depend on background orientation (*Figure 4B and H*, colored curves).

Our main interest here is in the target-evoked response in the presence of the background (*Figure 4C, I*), which can be extracted by subtracting the response to the background alone (*Figure 4A and G*) from the response to the target plus background (*Figure 4B and H*). If V1 contextual interactions at the retinotopic scale contribute to the behavioral effect of similarity masking, we would expect that target-evoked responses would be weakest with high target-background similarity (similar target and background orientations) and strongest with low target-background similarity (orthogonal target and background orientations).

Surprisingly, we find that the target-evoked response in V1 displays two distinct phases, with the early phase showing a paradoxical neural dependence on target-background orientation similarity that is anti-correlated with the behavioral masking effect, and with a later phase that is consistent with the behavioral masking effect. Specifically, in the early phase which starts at response onset, the target-evoked response is highest when the background matches the target orientation even though behaviorally this is the condition in which detection performance is the worst. However, after this initial phase, the high-similarity target-evoked response starts to drop, while the low-similarity target-evoked response continues to build up, so that in the later stages the target-evoked response is strongest on the dissimilar background and weakest on the similar background, consistent with the behavioral effect of similarity.

To quantify the relation between the effects of target-background orientation similarity on V1 population responses and behavior, we computed the correlation between the effect of orientation

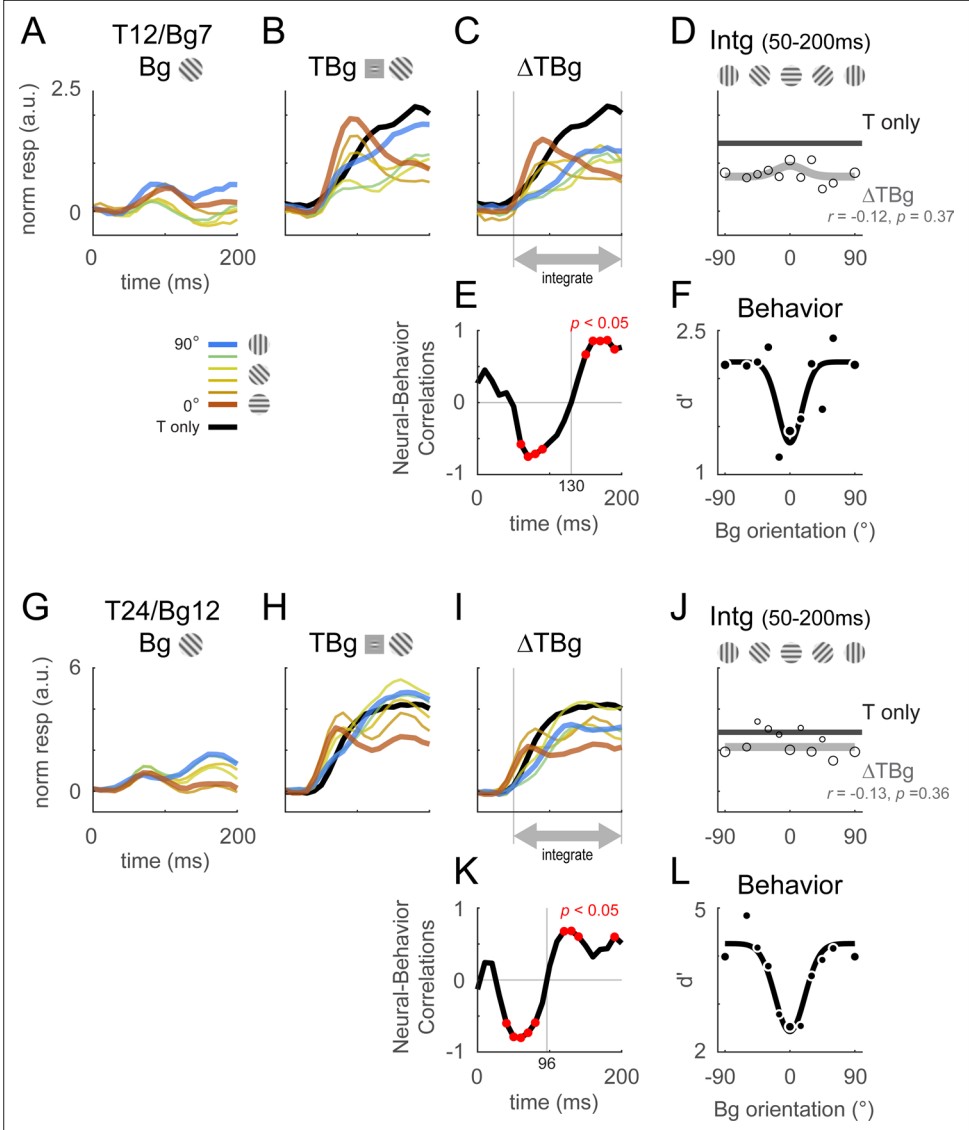

**Figure 4.** Retinotopic template dynamics and correlation to behavior. (**A–F**) Average response and dynamics from recordings with 12% target contrast (T12) and 7% background contrast (Bg7). (**A**) Response time course from stimulus onset (t=0ms) for background only trials. Background orientation are identified by color. Backgrounds with the same clockwise and anticlockwise orientation disparity from the target were pooled. (**B**) Response time course with the same additive target on different oriented background. Response of the target on uniform gray background is illustrated in black. (**C**) The target-evoked response time course was obtained by subtracting background only response (**A**) from response to target & background (**B**). Target evoked response was initially strongest for backgrounds close to 0° (red), then inverted around t=100ms such that response became the stronger for background closer to 90° (blue). (**D**) Response was averaged over 50–200ms and fitted with a Gaussian (gray) to illustrate the change in response magnitude with respect to background orientation. The neural-behavioral correlation of the response against behavioral response (**F**) is printed with $p$ significance value. Here, response to clockwise and anti-clockwise background orientations are plotted separately. Size of markers indicate the number of trials tested for each orientation. Black line indicates the response of the target only trials integrated over the same window. (**E**) The animals' behavior performance was anti-correlated with the initial phase of the retinotopic response, and was more aligned in the latter phase. Correlation coefficient was calculated across background orientations between each frame of the retinotopic response in (**C**) against the overall behavior performance in (**F**). Red dots indicate frames reaching statistical significance ($p<0.05$, $t$-test for correlation coefficient, see Methods). The neural-behavioral correlation crosses from negative to positive at t=130ms. (**F**) Behavior performance in d' was calculated as described in *Figure 2*. Size of markers indicate the number of trials tested for each orientation. Data was pooled across 8 experiments from both monkeys (see *Table 1*). (**G–L**) Same as (**A–F**) for

*Figure 4 continued on next page*

*Figure 4 continued*

recordings with 24% target contrast (T24) and 12% background contrast (Bg12). Similar trends were observed. The neural-behavioral correlation crosses from negative to positive at t=96ms in (**K**).

The online version of this article includes the following figure supplement(s) for figure 4:

**Figure supplement 1.** Retinotopic template dynamics and correlation to behavior for all combinations of background and target contrast levels.

**Figure supplement 2.** Retinotopic and columnar integrated response and correlation to behavior: correct trials vs all trials.

similarity on behavior (*Figure 4F and L*) and its effect on the target-evoked neural responses in individual 10ms frames (*Figure 4E and K*). This analysis reveals a robust paradoxical negative correlation between the early neural V1 response and behavior, weak positive correlation between the late neural V1 response and behavior, and no correlation between behavior and the integrated neural response. This result was obtained from averaging the response across all trials irrespective of whether the monkey made the correct decision.

To examined whether decision- and/or attention-related signals have a major contribution to the observed biphasic dynamics, we repeated the analysis on only the hits and correct rejection trials (*Figure 4—figure supplement 2B–C*). Our results are qualitatively the same for the subset of correct trials, indicating that decision- and/or attention-related signals are unlikely to play a major role in the observed dynamics.

Because the target and background are defined by their orientation, the correspondence between the neural signals in V1 and behavior may be better captured by V1 responses at the columnar scale. Our next step was therefore to examine the dynamics of the columnar-scale target-evoked responses in V1.

## Neural effects of target-background similarity masking at the scale of orientation columns

To study the effect of similarity masking on V1 responses at the columnar scale, we developed a linear columnar decoder of the VSDI signals (*Figure 3E*). The columnar decoder takes into account the location of the orientation columns within the retinotopic envelope of the target-evoked response. Because the target is horizontal, the output of the columnar template is expected to be positive for

**Table 1.** Experiment summary.

Experiment counts and the total number of trials included in the analysis presented in *Figures 2, 4, 5 and 7*; *Figure 4—figure supplements 1 and 2*; and *Figure 5—figure supplement 1*. Experiments with ineffective VSD staining were excluded from # Experiments. Trials with excessive motion or inconsistent EKG were excluded from # Total trials. Age of monkey reported at the time of the last listed experiment.

| | Target Contrast | Background Contrast | # Experiments | # Total Trials | # Hits | # Misses | # Correct Rejects (CR) | # False Alarms (FA) |
|---|---|---|---|---|---|---|---|---|
| **Monkey H** Male 8 years old | T12% | --- | 5 | 52 | 26 | 0 | 25 | 1 |
| | T12% | Bg7% | 3 | 593 | 214 | 80 | 253 | 46 |
| | T12% | Bg12% | 2 | 397 | 125 | 74 | 121 | 77 |
| | T12% | --- | 11 | 490 | 185 | 59 | 228 | 18 |
| | T12% | Bg7% | 5 | 1660 | 644 | 189 | 740 | 87 |
| | T12% | Bg12% | 6 | 1748 | 704 | 169 | 683 | 192 |
| | T24% | --- | 16 | 546 | 272 | 0 | 267 | 7 |
| | T24% | Bg7% | 5 | 1012 | 501 | 5 | 450 | 56 |
| | T24% | Bg12% | 6 | 1320 | 654 | 12 | 599 | 55 |
| **Monkey T** Male 7 years old | T24% | Bg24% | 4 | 388 | 204 | 2 | 155 | 27 |

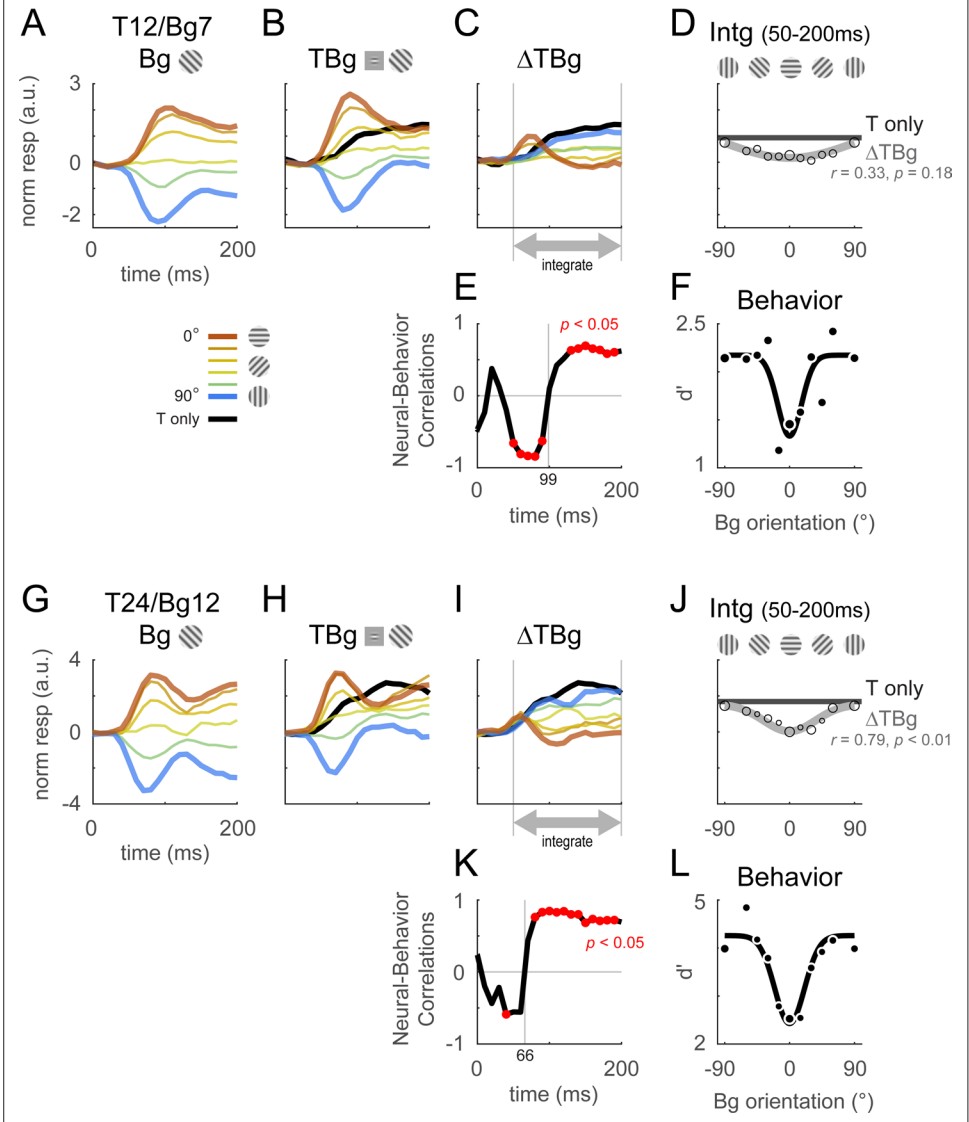

**Figure 5.** Columnar template dynamics and correlation to behavior. Same format as *Figure 4* with the response examined at the columnar scale. The biphasic response time course observed in the retinotopic scale was more pronounced at the columnar scale. (**A–F**) Averaged response and dynamics from recordings with 12% target contrast (T12) and 7% background contrast (Bg7). (**G–L**) Averaged response and dynamics from recordings with 24% target contrast (T24) and 12% background contrast (Bg12). Here, positive response represents relatively stronger activation of the neurons tuned to the target orientation (0°), and negative response represent stronger activation for neurons tuned to the orthogonal orientation (90°). Data pooling and counts are the same as reported in *Figure 4*. Behavioral correlation crosses from negative to positive at t=99ms in (**E**), and t=66ms in (**K**).

The online version of this article includes the following figure supplement(s) for figure 5:

**Figure supplement 1.** Columnar template dynamics and correlation to behavior for all combinatory background and target contrast levels.

the horizontal target and background stimuli and negative for the vertical background stimulus (since the horizontal and vertical columnar maps are anti-correlated).

As with the output of the retinotopic-scale template, the output of the columnar-scale template displays two distinct phases. *Figure 5* shows the time course of the columnar template signals to background alone (*Figure 5A and G*), the target plus background (*Figure 5B and H*), and target-evoked response in the presence of the background (*Figure 5C, I*). In the early phase, the target-evoked response is highest when the background and target have similar orientations, producing

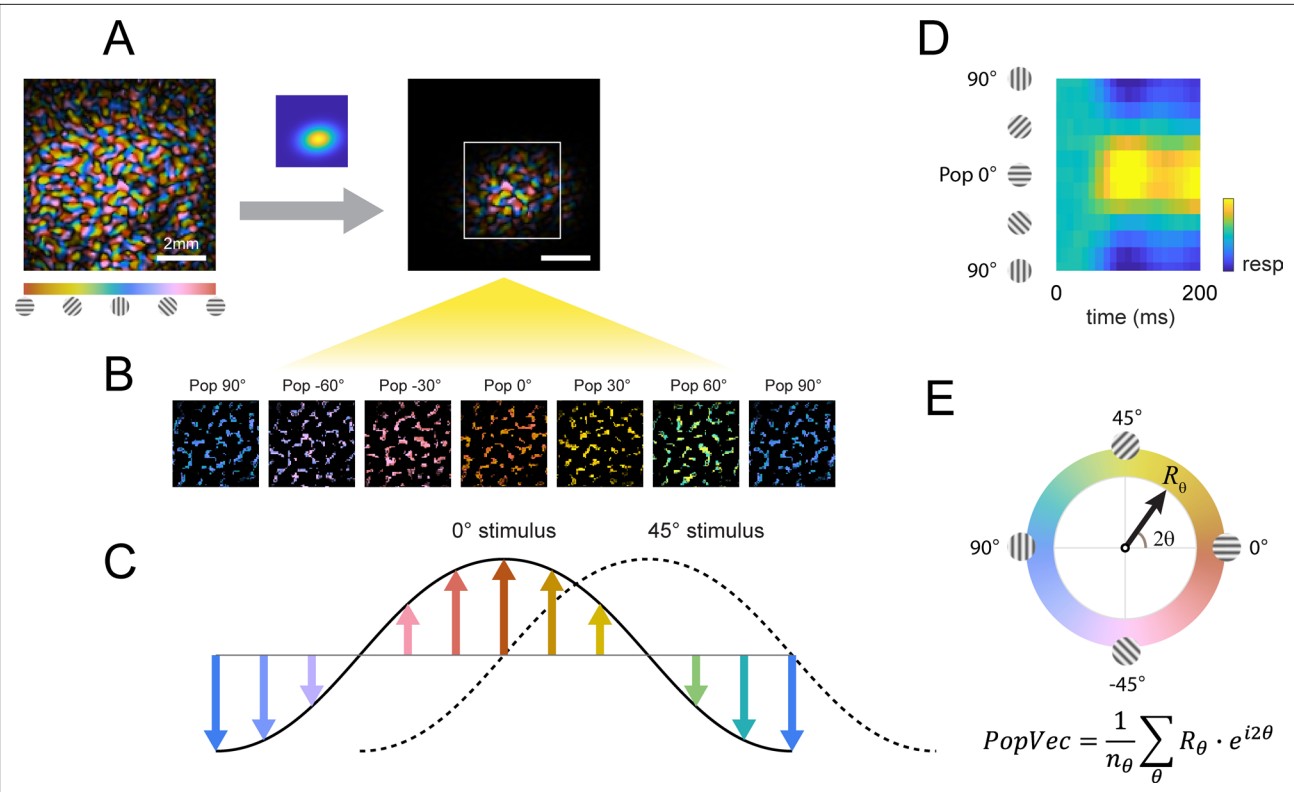

**Figure 6.** Columnar orientation estimation by populations tuning. (**A**) The orientation map obtained for each experiment as described in *Figure 3* was windowed to the retinotopic profile of the target. (**B**) Each pixel was assigned to one of 12 equally spaced orientation selective cluster maps by its preferred orientation. (**C**) The orientation selective decomposition of VSD response. To a grating stimulus oriented at 0°, the population tuning curve peaks at 0° (solid curve); likewise, the population peak would shift to 45° for a 45° grating (dotted curve). Note that this population response only represents the relative difference in preferred orientation (balanced positive and negative values); the overall neural response offset (retinotopic response) is not captured by this approach. (**D**) Example of a full population response time course from stimulus onset (t=0ms). (**E**) Population response can be summed to a complex vector representing the overall population tuning orientation and magnitude.

a paradoxical neural response that is anti-correlated with the behavioral masking effect (*Figure 5E and K*). In the second phase, the trend reverses and the target-evoked response is strongest on the dissimilar background and weakest on the similar background, consistent with the behavioral effect of similarity (*Figure 5F and L*). Similar results were obtained with other target and background contrast combinations (*Figure 5—figure supplement 1*).

Again, to examined whether decision- and/or attention-related signals have a major contribution to the observed biphasic dynamics at the columnar scale, we examined the behavioral correlations with hits and correct rejection trials only. We found only minor differences in the target-evoked response and behavioral correlations (*Figure 4—figure supplement 2D–E*), indicating that the observed biphasic dynamics at the columnar scale are unlikely to have a major top-down contribution.

Because the first phase of the response is shorter than the second phase, when V1 response is integrated over both phases, the overall response is positively correlated with the behavioral masking effect (*Figure 5D and J*, *Figure 5—figure supplement 1C*). Therefore, our results suggest that the neural masking effect at the columnar scale in V1 could play a major role in the behavioral similarity masking effects.

## Dynamics of columnar-scale orientation population trajectories

Our decoding analysis focuses on the columnar-scale orientation signals along the 0°–90° axis and reveals complex columnar-scale dynamic interactions between the target-evoked response and the response evoked by the background (*Figure 5*). To examine these dynamics in more detail, we performed two types of population-vector analyses (*Figure 6*). We began by assigning each pixel within the retinotopic footprint of the target-evoked response to one of 12 equally spaced preferred

orientations, creating 12 orientation selective clusters of pixels (*Figure 6B–C*). We then computed for each stimulus the response in each orientation selective cluster in each frame and displayed, in the first analysis, the population orientation tuning curve as a function of time (*Figure 6D*), and in the second analysis the population vector dynamic trajectory in the polar space spanned by the 12 orientations (*Figure 6E*); that is the orientation $\theta$ and magnitude $R_\theta$ of the peak of the population response over time.

The first population vector analysis reveals that V1 responses to the target or background alone are consistent with the stimulus orientation. In background only trials, shortly after stimulus onset the peak of the population tuning curve closely matches background orientation (*Figure 7A and F*, top row, red arrow and horizontal line). Similar results were obtained in the target only trials, where the peak of the population response tuning curve matches target orientation (*Figure 7A and F*, 4th row, green arrow and horizontal line).

Target plus background responses display complex spatiotemporal dynamics. To examine the dynamics of the target-evoked response in the presence of the background, we subtracted the background only response from the target plus background response (*Figure 7A and F*, 3rd row). The results reveal complex target-background interactions which could lead to a population tuning peak (white curve) that significantly deviates from the target orientation. For example, in some conditions, we observe an orientation tuning peak that is repelled from target orientation in the direction away from background orientation. An interesting goal of future studies would be to examine potential perceptual correlates of these interactions.

In the second population vector analysis, we plotted the response trajectories for each stimulus using the vector representation in polar coordinates (*Figure 7B–E and G–J*). After stimulus onset, the population vector for background only moves in the direction corresponding to the background orientation (*Figure 7B and G*) and for target only moves in direction corresponding to the target (*Figure 7E and J*).

The trajectories in the target plus background conditions are more complex. For example, when background orientation is at +/-45 deg to the target, the population response is initially dominated by the background, but then in mid-flight, the population response changes direction and turns toward the direction of the target orientation.

Such complex interactions can be used to constrain models of V1 population response.

## Dynamic gain control model qualitatively captures similarity masking effects in V1

Our next goal was to determine whether the observed interactions between the background- and target-evoked responses can be qualitatively captured by a gain control model. In this model, orientation columnar response was tuned to one of 12 equally spaced orientations. The responses of each orientation column were specified by the simple normalization model summarized in *Figure 8A*. Specifically, the spatiotemporal input stimulus generates an excitation signal and a normalization signal that are both linear with the input root mean square (rms) contrast. The normalization signal is then combined with a normalization constant to obtain the normalization factor. The normalized response is obtained by dividing the excitation signal by the normalization factor. The final response is then obtained by applying a response exponent $p$, which is similar to applying a spiking nonlinearity. Importantly, the excitation and normalization signals can differ in their spatial extent, orientation tuning width, and temporal impulse response (see Methods for model parameters).

We find that this simple model can qualitatively captures our key results. First, in response to background alone (*Figure 8B*), the modeled population vector peaked at ~100ms after stimulus onset and then dropped to a lower amplitude, as in our data (*Figure 5A and G*). This reduction in response amplitude was due to normalization signal that was delayed relative to the excitation signal. Second, as in the real data, response to the target plus background is less than the sum of the responses to each component separately. Third, as in our physiological results, the target-evoked response in the presence of the background is biphasic, having a brief early component in which the response is enhanced by target-background similarity, and a longer-lasting late component in which the response is suppressed by target-background similarity (*Figure 8D*). This leads to an early phase in which the response is anticorrelated with the behavioral effect of similarity masking, and a late phase and an

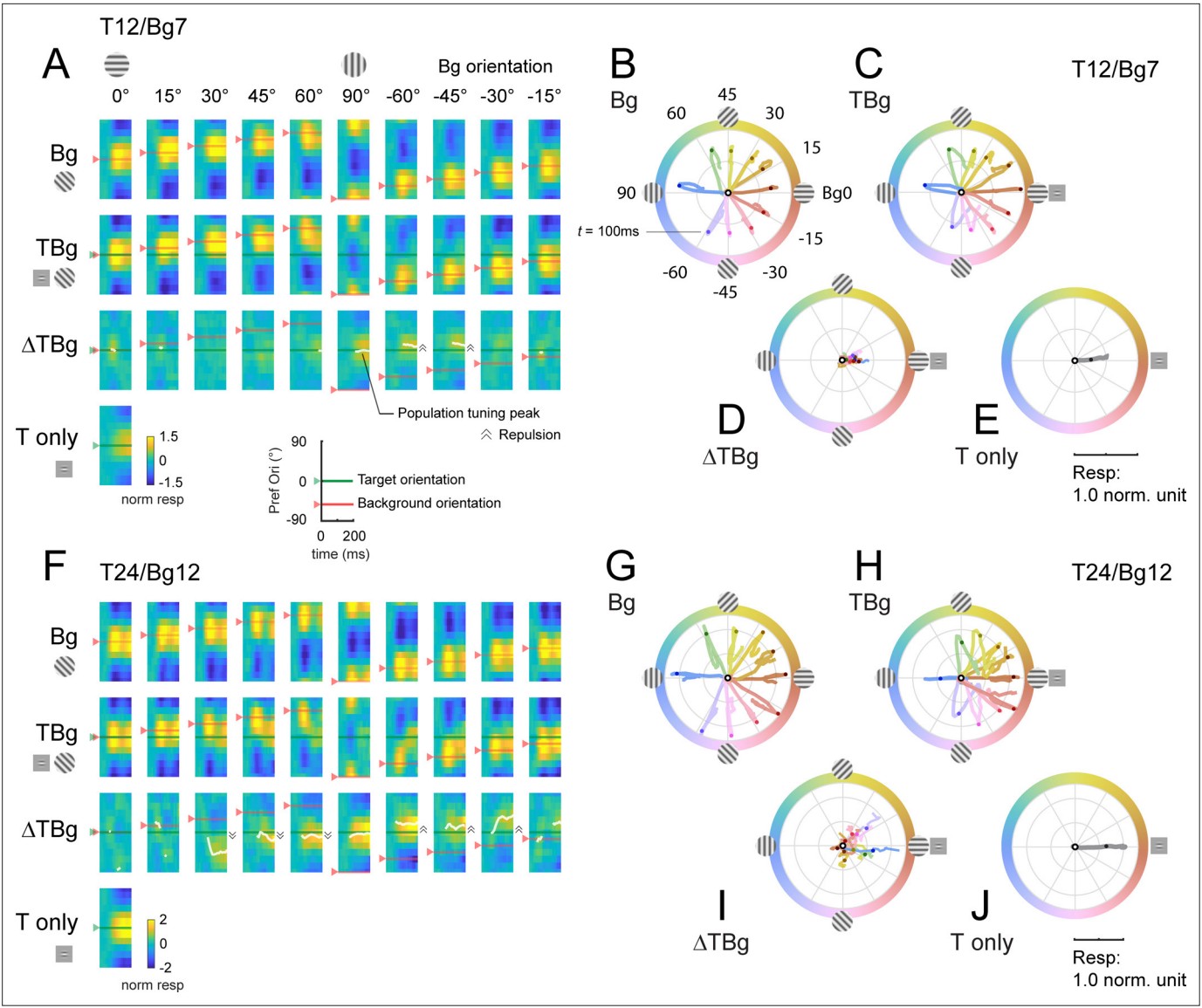

**Figure 7.** Dynamics of orientation population response. (**A**) Population tuning time course for trials with 12% contrast targets (T12) and 7% contrast backgrounds (Bg7). Averaged response time courses are presented as a heatmap with the y-axis representing the preferred orientation (from *Figure 6A–C*) and the x-axis time (see key on bottom right) to illustrate the change in the tuning over time. *Row 1:* Heatmaps for background only trials exhibit clear population tuning in the orientation of the background grating (red horizontal line). *Row 2:* Heatmaps for background with additive target showing population response dominated by the background orientation rather than the target orientation (0°). *Row 3:* The target evoked response is obtained by subtracting the background only response Row 1 from the target & background response in Row 2. Masking of the target evoked response was strong for backgrounds oriented near the target orientation (0°). With the background orthogonal to the target, population tuning in the target orientation can be identified. White line identifies the orientation of the population vector (peak tuning) wherever the normalized amplitude of the vector average was great than 0.2 (see Methods). Depending on background orientation, peak tuning appears to be offset from the orientation of the target (e.g. at Bg –45°). *Row 4:* Heatmap for the target only trials demonstrated clear population tuning in the target orientation (0°, green horizontal line). (**B–E**) Averaged response in (**A**) represented as a population vector form and illustrated as a continuous trajectory for each background orientation (color coded). (**B**) Population tuning trajectory for background only trials. The trajectories commenced in the center of the circle (white dot) and adhered closely to the orientation of the background. Dot on each trajectory indicates the position of the population tuning vector at 100ms. (**C**) Population tuning trajectory for background with additive target illustrating the biphasic response of this combined stimulus. In the early phase, the heading of the trajectory was a mixture of the background and target (0°) orientations, dominated more by the background. In the late phase, the trajectory made a sharp turn (t≈100ms) such that trajectories appeared to head towards a convergent point on the positive x-axis. (**D**) The trajectory for the target evoked response, calculated by subtracting the background only response (**B**) from the corresponding background & target (**C**). The target evoked response was weak and noisy, but was heading in the general direction of the target orientation (0°). (**E**) The population tuning trajectory for the target only trials illustrating clear tuning in the target orientation (0°). (**F–J**) Same as (**A–E**) for trials with 24% contrast targets (T24) and 12% contrast backgrounds (Bg12). Data was pooled and averaged across both monkeys.

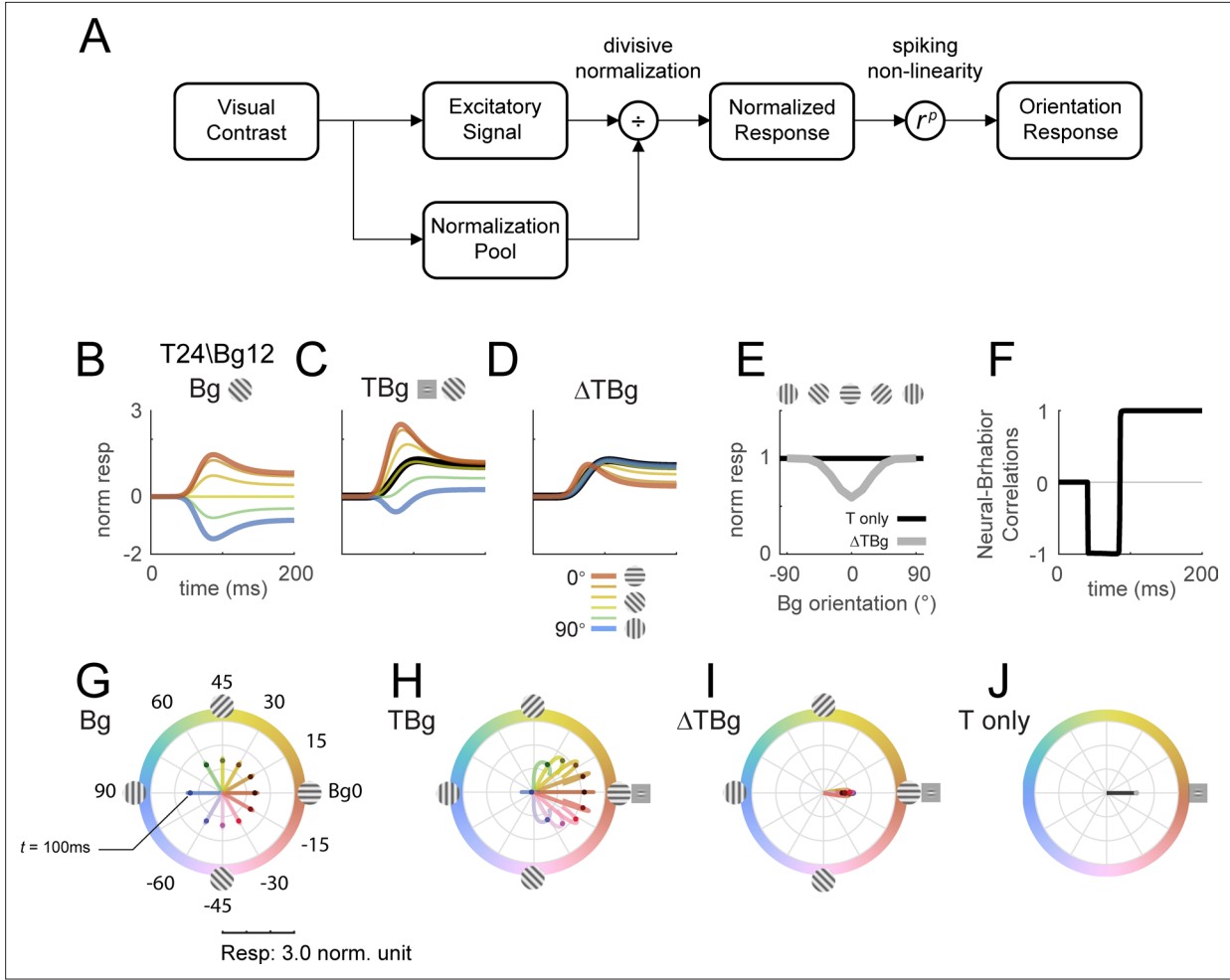

**Figure 8.** Delayed normalization model qualitatively captures key orientation masking response features. (**A**) Schematic of the normalization model showing the visual input being processed by separate excitatory and normalization signal pathways. The normalization pathway in particular was modeled with a slightly delayed temporal kinetics and wider orientation tuning curves. The excitatory signal undergoes divisive normalization prior to neural output. (**B–D**) Modeled columnar response output with target contrast of 24% and background contrast of 12%. Modeled response normalized to the target only response averaged over 50–200ms as plotted in (**E**). (**B**) Modeled response of oriented backgrounds as in *Figure 5A*. (**C**) Modeled response of background with additive target as in *Figure 5B*, and the modeled response of the small Gabor target in black. (**D**) The target evoked response was obtained by subtracting the background only response (**B**) from (**C**), matching the biphasic observation in *Figure 5C*. (**E**) Model response integrated over 50–200ms in the same format as *Figure 5D*. (**F**) Correlation of modeled behavioral performance (Gaussian fit in *Figure 5L*) against each time frame of the modeled response, illustrating the early phase where the response was negatively correlated to behavioral choice, and the late phase with positive correlations. (**G–J**) Modeled time course of the population tuning vector. (**G**) Modeled populating tuning trajectory of the background only stimuli (color coded) as in *Figure 7B*. (**H**) Modeled populating tuning trajectory of the background with additive target illustrating the turn towards a convergent point on the x-axis as in *Figure 7C*. (**I**) Modeled target evoked response trajectory from subtracting (**F**) from (**G**). (**J**) Modeled populating tuning trajectory of the target only trials as in *Figure 7E*.

The online version of this article includes the following figure supplement(s) for figure 8:

**Figure supplement 1.** Divisive normalization model with different normalization spatial extents.

**Figure supplement 2.** Divisive normalization model with different background spatial extents.

**Figure supplement 3.** Divisive normalization model with different background contrasts.

**Figure supplement 4.** Divisive normalization model with different normalization signal orientation tuning width.

integrated response that are positively correlated with the behavioral effect of similarity masking (*Figure 8E and F*).

Finally, this simple model can also display the curved trajectories of the population vector in response to the target plus background (compare *Figure 8H* to *Figure 7C and H*).

Additional results show that the model is relatively insensitive to the spatial extent of the normalization signal (*Figure 8—figure supplement 1*). The model predicts very similar temporal dynamics with the spatial extent of the background mask as small as twice the size of the target (*Figure 8—figure supplement 2*). When the background and target are the same size, the model predicts that sufficiently high contrast background will also drive the same biphasic temporal dynamics (*Figure 8—figure supplement 3*, rows 7 and 8). To account for the orientation-dependent neural and behavioral masking effects, the model requires an orientation tuned normalization (*Figure 8—figure supplement 4*).

Overall, our results suggest that a simple model with delayed and orientation tuned divisive gain control can qualitatively capture the complex spatiotemporal dynamics of V1 population responses to localized oriented targets added to oriented backgrounds.

## Discussion

To test the hypothesis that nonlinear computations in V1 contribute to the perceptual effect of similarity masking, we used voltage-sensitive dye imaging (VSDI) to measure neural population responses from V1 in two macaque monkeys while they performed a visual detection task in which a small oriented target was detected in the presence of a larger background of varied orientations. Like human observers, the monkeys were strongly affected by the orientation similarity of the target and the background (*Figure 2*). Their detection threshold increased with increased target-background orientation similarity, while their reaction times showed complex, and in some cases non-monotonic, dependency on target-background orientation similarity.

To quantify the neural effects of similarity masking, we measured neural sensitivity to the target at two fundamental spatial scales of V1 topographic representations. The large scale of the retinotopic map and the finer scale of the columnar orientation map. We discovered that at both scales, V1 population responses to the target and background display two distinct phases (*Figure 4B and H*, *Figure 5B and H*). An initial transient phase in which target-evoked V1 response is strongest when the target and background have similar orientations. At this early phase, V1 responses are therefore paradoxically anti-correlated with the behavioral effect of similarity masking (*Figure 4E and K*, *Figure 5E and K*). In the second phase, the masking effect reverses, and the target-evoked response is maximally reduced when the target and background are similar. In this second sustained phase, V1 population responses are therefore consistent with the behavioral similarity masking effect. To explore the possibility that these biphasic dynamics reflect contributions from decision- and/or attention-related top-down signals rather than from low-level nonlinear encoding mechanisms in V1, we re-examined our results while excluding error trials (*Figure 4—figure supplement 2*). We found that the biphasic dynamics hold even for the subset of correct trials, reducing the likelihood that decision/attention-related signals play a major role in explaining our results.

The positive correlation between the neural and behavioral masking effects occurred earlier (*Figure 5E and K* vs. *Figure 4E and K*) and was more robust at the columnar scale than at the retinotopic scale (*Figure 5D and J* vs. *Figure 4D and J*; see also *Figure 4—figure supplement 1C*, and *Figure 5—figure supplement 1C*). In addition, while the temporally integrated columnar response was positively correlated with behavior across all target and background contrasts tested (*Figure 5E and K*, *Figure 5—figure supplement 1D*), the integrated retinotopic responses were uncorrelated, or in some cases anticorrelated, with behavior (*Figure 4E and K*, *Figure 4—figure supplement 1D*). These results suggest that behavioral performance in our task is dominated by columnar scale V1 signals in the second phase of the response. To the best of our knowledge, this is the first demonstration of such decoupling between V1 responses at the retinotopic and columnar scales, and the first demonstration that columnar scale signals are a better predictor of behavioral performance in a detection task.

Due to the challenges of setting up these experiments, we were unable to collect all target/background contrast combinations from both monkeys. However, in the common conditions, the results appear similar in the two animals, and the key results seem to be robust to the contrast combination in the animal where a wider range of contrast combinations was tested (*Figure 4—figure supplement 1*, and *Figure 5—figure supplement 1*).

We find that when the target and background have similar orientations, columnar-scale information about the target is restricted to the first phase of the response and then largely disappears during the

second phase of the response. These physiological results could be related to the surprising mismatch between task difficulty and reaction times (*Figure 2C*). Rather than having reaction times that monotonically increase with task difficulty, in our masking detection task, reaction times can be shortest when target and background orientations match, even though it is hardest to detect the target under these conditions. The short reaction time to this stimulus may be the consequence of the target information being best represented in the early phase of the response.

The nature of contextual modulations in V1 is quite complex (*Angelucci et al., 2017*; *Angelucci and Bressloff, 2006*; *Bai et al., 2021*; *Cavanaugh et al., 2002b*; *Henry et al., 2020*; *Michel et al., 2018*; *Polat et al., 1998*; *Sceniak et al., 1999*; *Shushruth et al., 2012*). A second goal of our study was to quantitatively characterize the spatiotemporal dynamics of columnar-scale V1 population responses to targets and backgrounds of different orientations and contrasts. Using a dynamic population vector analysis, we find that in the presence of an oriented background, the peak of the population orientation tuning to the target can deviate significantly from target orientation. For example, in some conditions, we observe a population orientation tuning peak that is repelled away from target orientation in the direction opposite to background orientation (*Figure 7E*). These orientation-dependent interactions could contribute to non-veridical perceptual representations of orientation such as in the well-known tilt illusion effect (*Clifford, 2014*; *Schwartz et al., 2007*; *Wenderoth and Johnstone, 1987*). An important goal for future studies would be to test for this possibility.

Using the population vector analysis, we find that columnar scale V1 representations are initially dominated by the orientation of the background. The target orientation then appears in the second phase of the response, which leads to curved population vector trajectories (*Figure 7C and H*). Identifying possible perceptual consequences of such dynamic and complex trajectories, and understanding the neural circuit mechanisms that give rise to such responses, are two important goals for future work.

Nonlinear response properties in V1 are commonly modeled as a consequence of a divisive gain control mechanism (*Albrecht and Geisler, 1991*; *Carandini and Heeger, 1994*; *Heeger, 1991*; *Heeger, 1992*; *Sit et al., 2009*). As a first step toward understanding the mechanisms that could give rise to the observed V1 responses, we tested whether a simple dynamic gain control model could account for our findings (*Figure 8*). We find that a simple gain control model can qualitatively account for our results, but that in order to do so, the model has to display two important properties. First, to account for the biphasic nature of V1 response, the divisive normalization signals have to be delayed relative to the excitatory signal. Second, in order to account for the reduced neural sensitivity with target-background similarity in the second phase of the response, the divisive normalization signal has to be orientation selective (*Figure 8—figure supplement 4*). Because in primates and carnivores, robust orientation selectivity first emerges in V1 (*Hubel and Wiesel, 1959*; *Hubel and Wiesel, 1968*), these results suggest that a significant portion of the nonlinear interactions observed in the current study originate in V1 rather than being inherited from the ascending inputs that V1 receives from the LGN. While our experimental and computational results point to a delayed gain control signal that operates at the level of V1, they do not directly speak to the circuit and biophysical mechanisms that contribute to the implementation of this gain control in V1. Multiple candidate mechanisms for implementing gain control in V1 have been proposed (*Angelucci et al., 2017*; *Angelucci and Bressloff, 2006*; *Ozeki et al., 2009*; *Rubin et al., 2015*; *Tsodyks et al., 1997*). Our results provide new and powerful constraints for such mechanistic models.

A key difference between our study and previous center-surround studies (e.g. *Cavanaugh et al., 2002a*; *Cavanaugh et al., 2002b*; *Henry et al., 2020*; *Shushruth et al., 2012*) is the stimuli that we used. First, in our experiments, the target and the mask were additive, while in most previous center-surround studies the target occludes the background. Such studies therefore restrict the mask to the surround while our study allows target-mask interactions at the center. Second, most previous center-surround studies have a sharp-edged target/surround border, while in our experiments no sharp edges were present. Unpublished results from our lab suggest that such sharp edges have a large impact on V1 population responses. Third, our stimuli were flashed for a short interval of 250ms corresponding to a typical duration of a fixation in natural vision, while most previous center-surround studies used either longer-duration drifting stimuli or very short-duration random-order stimuli for reverse-correlation analysis.

Because our targets are added to the background rather than occluding it, it is likely that a significant portion of the behavioral and neural masking effects that we observe come from target-mask interactions at the target location rather than from the effect of the mask in the surround. Several lines of evidence support this possibility. First, in human subjects, perceptual similarity masking effects can be almost entirely accounted for by target-mask interactions at the target location and are recapitulated when the mask has the same size and location as the target (*Sebastian et al., 2017*). There is a reduction in masking when the background is windowed to the target envelope, but this effect is due to removing background within the target envelope (*Sebastian et al., 2017*). Second, in our computational model (*Figure 8*), the effects of mask orientation on the dynamics of the response are qualitatively similar if the mask is restricted to the size and location of the target and mask contrast is increased (*Figure 8—figure supplement 3*). Third, in our model, the results are qualitatively the same when the spatial pooling region for the normalization signal is the same as that for the excitation signal (*Figure 8—figure supplement 1*). These considerations suggest that center-surround interactions may not be necessary for neural and behavioral masking effects with additive targets.

Finally, we note that the tuned similarity normalization that explains the neural and behavioral similarity-masking effects reported here is consistent with a principled encoding strategy for feature detection under natural conditions. When viewing a static scene under natural conditions, human and non-human primates make 3–4 saccadic eye movements per second, with fixations between saccades of 200–300ms. Given the typical size of the saccades, most visual receptive fields are stimulated during each fixation by a largely statistically independent random sample of natural image (*Frazor and Geisler, 2006*). Analysis of the responses of linear receptive fields to random samples of natural image shows that the standard deviation of the response increases in proportion to the product of the luminance, the contrast and the similarity of the natural background to the receptive field (*Sebastian et al., 2017*). Thus, divisive normalization by the product of luminance, contrast and similarity causes the standard deviation of the responses across natural images to be much more constant (i.e. nearly independent of the luminance, contrast and similarity of the background within the receptive field). This more constant standard deviation makes it possible, with relatively simple decoders, to reach near optimal feature-detection performance, under the high levels of stimulus uncertainty that occur under natural conditions (*Schwartz and Simoncelli, 2001*; *Sebastian et al., 2017*).

## Methods

All procedures have been approved by the University of Texas Institutional Animal Care (IACUC protocol #AUP-2016–00274) and Use Committee and conform to NIH standards.

**Key resources table**

| Reagent type (species) or resource | Designation | Source or reference | Identifiers | Additional information |
|---|---|---|---|---|
| Voltage-sensitive dye | RH1691; RH1838 | Optical Imaging Inc. | RH1691; RH1838 | |

### Widefield voltage-sensitive dye imaging

The experimental technique for widefield voltage-sensitive dye (VSD) imaging of neural response in awake, behaving macaques was adapted from previous studies (*Bai et al., 2021*; *Chen et al., 2006*; *Chen et al., 2008*; *Chen et al., 2012*). Briefly, two adult male rhesus macaque monkeys (Monkey H, 8 years old; and Monkey T, 7 years old) were implanted with a metal head post and metal recording chambers located over the dorsal portion of V1, a region representing the lower contralateral visual field at eccentricities of 2–5°. Craniotomy and durotomy were performed. A transparent artificial dura made of silicone was used to protect the brain while allowing optical access for imaging (*Arieli et al., 2002*; *Figure 3A*). Experiments were conducted in the left hemisphere chamber of Monkey H, and in both left and right hemisphere chambers of Monkey T.

VSD imaging was used to record neural population activity at a high resolution in space and time (*Shoham et al., 1999*). Before each experiment, VSD (RH1691 or RH1838, Optical Imaging, Inc) was topically applied on the cortex for 2 hr to allow the VSD molecules to bind to cellular membranes. In Monkey H, fluorescence from neural activity was recorded using Imager 3001 (Optical Imaging, Inc) using a tungsten-halogen light source (Zeiss). An infrared eye-tracker (Dr Bouis Inc) was used to

monitor eye position. In Monkey T, florescence was recorded using custom Matlab software interfaced to PCO Edge 4.2 sCMOS camera (Excelitas PCO GmbH) using X-Cite 110LED light source (Excelitas Technologies Corp). Eye position was monitored using an Eyelink 100 Plus video eye-tracker (SR Research Ltd).

Both imaging systems were interfaced to a double-SLR-lens-macro system with housing for dichroic mirrors in between the two SLR lenses. The combination of a 50 mm fixed-focus objective lens (cortex end, Nikkor 50 mm f/1.2) and an 85 mm fixed-focused (Canon EF 85 mm f/1.2 L USM) camera lens provided 1.7 x magnification, corresponding to imaging approximately an 8x8 mm$^2$ area of the cortex. Fluorescence signals were measured through a dichroic mirror (650 nm long-pass filter) and an emission filter (RG 665). VSD molecules were excited by light at 630 nm. Imaging data were collected at 512×512 resolution at 100 Hz. Data acquisition was time locked to the animal's heartbeat (EKG QR up-stroke, HP Patient Monitor HP78352C). More details about optical imaging with VSD in behaving monkeys are described elsewhere (*Bai et al., 2021*; *Chen et al., 2006*; *Chen et al., 2008*; *Chen et al., 2012*).

Prior to the main experiments, VSD imaging was used to obtain a precise retinotopic map of the entire recording (*Figure 3—figure supplement 1*; *Yang et al., 2007*). In two out of three chambers, retinotopic maps indicate that V1 extended into the lunate sulcus. In the third chamber, V1 terminated ~0.75 mm from the lunate sulcus. The area used for decoding analysis was chosen to entirely lie within V1.

## Behavioral task with optical stimulation

Monkeys were trained to detect a small additive horizontal Gabor target (4cpd, with σ=0.14°, 0.33° FWHM envelope) centered on a sinusoidal grating background mask of the same spatial frequency (4° raised-cosine windowed). The background grating was oriented at 0°, ±15°, ±30°, ±45°, ±60, and 90° from the Gabor target orientation. Both the Gabor and background grating were bright-centered – that is, the 0° orientation background was completely in phase with the target. The contrast of the target and background were varied in combinations of levels, reported in Michelson contrast:

$$c = \frac{L_{max} - L_{min}}{L_{max} + L_{min}} = \frac{L_{max} - L_{background}}{L_{background}}$$

For each experiment, a Fixation recording block and a Detection recording block were made using the same target and background conditions. In both blocks, the target and background were centered at a fixed position for each experiment corresponding to the working cortical chamber. This position varied between experiments from 1.6 to 3 deg of visual angle eccentricity from the fixation point and from 20 to 50 deg of polar angle from the vertical meridian in the corresponding hemifield (*Figure 3— figure supplement 1*). At these coordinates, the spatial extent of the target was fully imaged through the cortical window, and the larger oriented background uniformly activated the entire imaging area.

In the Fixation block, the monkeys were required to remain fixated for each imaging trial while either the target or full-field sinusoidal gratings at 100% contrast were flashed at 5 Hz (60ms ON, 140ms OFF) for 1.0 s. These recordings were processed to obtain retinotopic and columnar orientation response maps that were used to decode the detection recording responses from the same experiment day (*Figure 3C–E*).

In the Detection blocks, a background with random orientation appeared on every trial, with a 50% chance of an accompanying Gabor target (*Figure 1C–E*). The monkeys were tasked to report the presence of the target. Each trial began with fixation on a bright 0.1° square. An auditory tone and the dimming of the fixation square cued the monkey to the start of the detection task trial. 250ms later, the background with or without the target was presented. The monkeys were trained to maintain gaze at the fixation cue on target absent trials or saccade to and hold gaze (for 150ms) at the target position to indicate target detection (with a 75ms minimum allowed reaction time). When the target was present, it remained on screen for a maximum of 250ms or was extinguished immediately upon the monkeys' saccade initiation. The monkey was given 600ms to make the saccade or to hold fixation and was subsequently rewarded on correct choices: stay (correct reject) on target absent trials, or saccade to target (hit) on target present trials. The target and background contrast level were fixed for each recording block. The probability of each orientated background and of target presence were balanced for each recording block.

A separate target only Detection block on uniform gray background was also taken on each experiment day using the same routine as above. This block was later used as the reference data to normalize response amplitudes across experiment days.

Experiments were conducted with custom code using TEMPO real-time control system (Reflective Computing). The visual stimulus was presented on a Sony CRT (1024x768 @ 100 Hz), distanced 108 cm from the animal (50 pixels-per-degree), with mean luminance 50 cd/m². The visual stimulus was generated using in-house real-time graphics software (glib).

## Behavior performance and reaction time

Behavior performance was calculated for each background orientation and reported in units of detection sensitivity index d' (d-prime). D-prime and criterion were estimated as:

$$d' = \Phi^{-1}\left(P(HIT)\right) - \Phi^{-1}\left(P(FA)\right), \qquad criterion = -\frac{\Phi^{-1}\left(P(HIT)\right) + \Phi^{-1}\left(P(FA)\right)}{2},$$

where $\Phi^{-1}(x)$ is the inverse transform of the cumulative normal distribution; and $P(HIT)$ and $P(FA)$ represent the proportion of hits and false alarms respectively. To avoid leaving out the data in some conditions (e.g., when there are no false alarms), we scaled all the proportions to be between 0.005 and 0.995 ($P(x) = 0.005 + 0.99 \cdot P(x)$).

Mapping between the unbiased percentage correct response and d' is as follows and is depicted in *Figure 2B*:

$$d' = 2 \cdot \Phi^{-1}(PC_{max}); \qquad PC_{max} = \Phi\left(\frac{d'}{2}\right)$$

D-prime performance across orientations was fitted with an inverted, dc-shifted Gaussian:

$$y = Ae^{\frac{-\theta^2}{2\sigma^2}} + c$$

The reaction times of the animals were calculated from the stimulus onset to the onset of saccade. Consequently, there was no measurement of reaction time on trials where the animals remain fixated at the fixation point.

## VSD imaging

For each trial, an image sequence was captured for a total of 1.2 s including pre-stimulus and post-stimulus frames. The image sequence was analyzed to extract the response using a variant of the previous reported routines (*Bai et al., 2021*; *Chen et al., 2006*; *Chen et al., 2008*; *Chen et al., 2012*).

Image stabilization was introduced as the first stage of pre-processing to de-accentuate blood vessel edges in the ΔF/F response map caused by micro movements of the camera and/or the cortex during imaging. The image intensity across time at each individual pixel was modeled with separable motion-free ($I_{x0}(t)$) and motion-related ($\vec{\alpha}_x . \vec{v}(t)$) components as follows:

$$I_x(t) = I_{x0}(t) + \vec{\alpha}_x \vec{v}(t)$$

For each trial, a single global motion vector $\vec{v}(t)$ was obtained by estimating the translational motion of the center portion of the images (1/4 of the imaging area). The motion coefficients $\vec{\alpha}_x$ for each pixel was then obtained using least squares fitting to the model. The motion-corrected image is $I_{x0}(t)$. This approach to image stabilization (compared to traditional image registration approach) has the advantage of correcting for non-rigid movements (rotations, expansion/contractions, affine transformation, local distortions, etc.) and sub-pixel motion.

## Retinotopic and columnar template decoding

Template decoding was used to summarize the retinotopic and columnar response for each image frame. The retinotopic response map of the target and columnar orientation map of the imaging area were estimated from the Fixation blocks, in which response were stimulated with visual presentation

at 5 Hz. The preprocessing steps for the Fixation blocks were: image stabilization, 5 Hz FFT response extraction, ΔF/F normalization, then down-sampling to 128x128 pixels (from 512x512). Baseline florescence ($F_0$) was estimated by the average florescence over frames –80 to 0ms relative to stimulus onset. ΔF/F was calculated as:

$$\frac{\Delta F}{F}(t) = \frac{F(t) - F_0}{F_0}$$

The retinotopic response map of the Gabor target ($H_{ret}(x)$) was estimated by fitting a 2D Gaussian over the 5 Hz flashing Gabor amplitude response. The full-ROI (8x8 mm$^2$ imaging area) columnar orientation response maps ($H_{ori}(x)$) were estimated from the flashing full-field grating, where the 5 Hz FFT grating response amplitudes were bandpass filtered from 0.8 to 3.0 cycles/mm and the orientation tuning of each pixel was estimated as described previously (*Chen et al., 2012*). Subsequently, the full-ROI orientation map was windowed by the retinotopic map to co-localize the retinotopic and the columnar decoders. The columnar map, comprised of pixelwise response magnitude ($A(x)$) and tuning angle ($\theta(x)$), was represented in modified Euler's form:

$$H_{ori}(x) = H_{ret}(x) A(x) e^{2i\theta(x)}$$

These maps served as templates for decoding the retinotopic and columnar response from the Detection blocks. To further reduce the effect of motion artefacts, a pixel-wise reliability weighted approach was adopted. The Detection blocks were preprocessed with image stabilization, down-sampling to 128x128, and ΔF/F normalization as above. From the pre-processed images, the retinotopic scale variance $\sigma^2_{ret}(x)$ and columnar scale variance $\sigma^2_{col}(x)$ were calculated to be used as reliability weights. The variance was obtained from condition-mean subtracted residuals taken across all frames across trials, with the ΔF/F response ($R_{ret}(x,t)$) bandpass filtered between 0.8 and 3.0 cycles/mm ($R_{col}(x,t)$) for the columnar scale variance. Reliability weighting was implemented by normalizing each template pixel by the corresponding pixel-wise variance.

$$\hat{H}(x) = \frac{H(x)}{\sigma^2(x)}$$

Two different columnar decoding methods were employed. The first examined the overall response aligned to the orientation of the Gabor (0° orientation tuning axis). A second columnar decoding scheme was employed to examine the full orientation population response. This second scheme was comprised of 12 decoders that evenly partitioned the orientation space, such that each decoder contained the column response magnitude in a subset of pixels tuned with ±7.5° centered at –75° to 90° in 15° steps. Each decoder therefore represents the population response of similarly tuned neural ensembles spanning the orientation space every 15°. Each decoder was normalized by the summed response magnitude of its pixel subset; in this way, each sub-population response was equally represented in the population tuning curve.

The formulation for the reliability-weighted templates and the decoding is summarized below for the retinotopic response time course ($r_{ret}(t)$), columnar response time course ($r_{0-90}(t)$), columnar population tuning time course ($r_\theta(t)$).

$$r_{ret}(t) = R_{ret}(\mathbf{x},t) \frac{\hat{H}_{ret}(\mathbf{x})}{\left\| \hat{H}_{ret} \right\|}, \qquad \hat{H}_{ret}(x) = \frac{H_{ret}(\mathbf{x})}{\sigma^2_{ret}(\mathbf{x})}$$

$$r_{0-90}(t) = R_{col}(\mathbf{x},t) \frac{\hat{H}_{0-90}(\mathbf{x})}{\left\| \hat{H}_{0-90} \right\|}, \qquad \hat{H}_{0-90}(x) = \frac{real\left(H_{ori}(\mathbf{x})\right)}{\sigma^2_{col}(\mathbf{x})}$$

$$r_\theta(t) = R_{col}(\mathbf{x},t) \frac{\hat{H}_\theta(\mathbf{x})}{\left\| \hat{H}_\theta \right\|}, \qquad \hat{H}_\theta(\mathbf{x}) = \frac{H_\theta(\mathbf{x})}{\sigma^2_{col}(\mathbf{x})'}, \qquad H_\theta(\mathbf{x}) = \frac{\left| H_{ori}(\mathbf{x}|\theta) \right|}{\sum_x \left| H_{ori}(\mathbf{x}|\theta) \right|}$$

## Response pooling across experiments

Across experiments, the varying effectiveness of VSD staining led to large variations in the noise level and amplitude of the ΔF/F response. For pooling data across experiments, response amplitudes were normalized based on singular value decomposition (SVD). Here, the decoded response of the reference target only Detection block was used. These were trials of varying target contrast levels on the uniform, mean luminance background instead of the oriented grating background, collected on the same day. The assumption is that the neural response amplitude and dynamics to the target should be the same irrespective of the experiment day. Retinotopic and columnar decoder responses for the target only block were calculated as with the background detection block as described above. Target contrast response for each experiment were interpolated so that all assessed target contrast levels across experiments were represented, then SVD was performed over the frames –100 to 200ms about the stimulus onset across all experiments. In this way, the first component of the SVD (SVD1) represented the average neural dynamics of the target contrast response, and the SVD1 coefficients represent the magnitude of this target contrast response each experiment day. Response from each experiment was then scaled to match the experiment with the largest SVD1 coefficient. This was performed for each decoded response separately. When pooling data for the mean and standard deviation, the inverse of the scaling factor squared was used as the reliability weighting.

$$\bar{r} = \frac{1}{n} \sum_{exp,trial} w_{exp} r_{exp,trial}, \qquad \sigma_r = \sqrt{\frac{1}{n} \sum_{exp,trial} \left( w_{exp} r_{exp,trial} - \bar{r} \right)^2},$$

$$w_{exp} = \frac{1}{scaler_{exp}^2} = \frac{1}{\left( SVD1_{max} / SVD1_{exp} \right)^2}$$

Lastly, responses are expressed in a modified z-score, obtained by normalizing the response by its standard deviation. To calculate this standard deviation, responses were grouped by presented stimuli, and the means of each group was subtracted. The residuals from the mean-subtraction from frames 50–250ms post stimulus onset was pooled according to the aforementioned experiment reliability weights $w_{exp}$ to obtain the response standard deviation.

Response beyond the saccade may contain unwanted signals. Frames beyond the reaction time for each trial were therefore omitted from summary statistics. For the integrating response within trials, responses were averaged up to the frame of saccade. For frame-by-frame averaging across trials for response time course, trials were dropped out from the averaging beyond their reaction time frame.

## Descriptive trends across background orientation

Trends were fitted to the normalized VSD response across background orientations (e.g. gray curve in *Figure 4D*). VSD responses were first averaged over a specific range of frames, and for illustration only the trends of the averaged response across orientations were fitted with either a flat line or a Gaussian.

$$\widetilde{r}_{flat} = m\theta + c$$

$$\widetilde{r}_{gau} = Ae^{\frac{-\theta}{2\sigma^2}} + c$$

The best fitting trend by the F-test were chosen for display in the figures.

$$\frac{\frac{SS_2 - SS_1}{df_2 - df_1}}{\frac{SS_1}{df_1}} F\left( df_1, df_2 - df_1 \right)$$

## Behavior correlation

The retinotopic and the columnar response time courses were correlated against the monkey's behavior judgement. Pearson's correlation coefficient was estimated between the instantaneous response in time and overall behavior sensitivity index $d'$. To account for the different trial counts between background orientations, a trial count weighted version of the correlation coefficient was adopted:

$$\rho = \frac{cov\,(x, y; w)}{\sqrt{cov\,(x, x; w)\, cov\,(y, y; w)}}$$

$$cov(x, y; w) = \frac{\sum \left( w_i(x_i - m(x; w)) \right) \left( w_i \left( y_i - m(y; w) \right) \right)}{\sum w_i}, \qquad m\,(x; w) = \frac{\sum w_i x_i}{\sum w_i},$$

The p-value for the weighted coefficients were estimated using the standard t-score replacing the degree of freedom with an entropy based effective estimate from the weights.

$$t_\rho\left(\rho; w\right) = \frac{\rho \sqrt{n_{eff} - 2}}{\sqrt{1 - \rho^2}} \sim tcdf\left(n_{eff} - 2\right)$$

$$n_{eff} = exp(H) = exp\left(\Sigma n_{wi} ln(n_{wi})\right), n_{wi} = w_i/\Sigma w_i,$$

## Normalization model of orientation masking dynamics

A simple model of the neuronal population response with divisive normalization described our results qualitatively. In this model, orientation columnar response was tuned to one of 12 different orientations: –75° to 90° in 15° increments. The responses of each orientation column were specified by the simple normalization model summarized in **Figure 8**. The input stimulus is specified by the contrast and orientation of the target, the contrast and orientation of the background, and the duration of the stimulus: $\mathbf{s} = (C_T, \theta_T, C_B, \theta_B, D)$. In the model, the input stimulus generates an excitation signal $r_e\left(t|\mathbf{s}, \theta_{max}\right)$ that is linear with stimulus contrast and a normalization signal $r_n\left(t|\mathbf{s}, \theta_{max}\right)$ that is also linear with stimulus contrast. Without loss of generality, all signals were scaled by the 24% contrast target only response averaged over 50–200ms. These excitation and normalization signals are controlled by the excitation and normalization parameters, $\mathbf{\Omega}_e$ and $\mathbf{\Omega}_n$, described below. The normalization signal is then combined with a normalization constant $r_0$ to obtain the normalization factor. The normalization constant limits how small the normalization factor can become. The normalized response is obtained by dividing the excitation signal by the normalization factor. The final response is then obtained by applying a response exponent $p$, which is similar to applying a spiking nonlinearity:

$$r\left(t|\mathbf{s}, \theta_{max}\right) = \left[\frac{r_e\left(t|\mathbf{s}, \theta_{max}\right)}{normalization\ factor}\right]^p = \left[\frac{r_e\left(t|\mathbf{s}, \theta_{max}\right)}{\sqrt[p]{r_n^p\left(t|\mathbf{s}, \theta_{max}\right) + r_0^p}}\right]^p = \frac{r_e^p\left(t|\mathbf{s}, \theta_{max}\right)}{r_n^p\left(t|\mathbf{s}, \theta_{max}\right) + r_0^p}$$

In **Figure 8**, we show the final responses corresponding to the center of the stimulus $\mathbf{x}_0$. The excitatory response at that location for background, and target plus background, is obtained by convolving the effective input contrast signals with the spatial-temporal impulse-response function and evaluating at $\mathbf{x}_0$

$$r_e\left(t|\mathbf{s}, \theta_{max}\right) = \begin{cases} \left(c_{Be} * h_e\right)\left(\mathbf{x}_0, t\right) \\ \left(c_{TBe} * h_e\right)\left(\mathbf{x}_0, t\right) \end{cases}$$

where $c_{Be}\left(\mathbf{x}, t\right)$ and $c_{TBe}\left(\mathbf{x}, t\right)$ are the effective input contrast signals, and $h_e\left(\mathbf{x}, t\right)$ is the spatiotemporal impulse response function. The effective excitatory contrast of the background for the orientation channel with preferred orientation $\theta_{max}$ is given by

$$c_{Be}\left(\mathbf{x}, t\right) = C_B\left(\mathbf{x}\right) exp\left(-\frac{\left(\theta_B - \theta_{max}\right)^2}{2\sigma_e^2}\right) w\left(t; D\right)$$

where $C_B\left(\mathbf{x}\right)$ is the background contrast, $\sigma_e$ is the falloff parameter of column's orientation turning function, and $w\left(t; D\right)$ is a temporal pulse function of width $D$. Similarly, the effective target contrast is given by

$$c_{Te}\left(\mathbf{x}, t\right) = C_T\left(\mathbf{x}\right) exp\left(-\frac{\left(\theta_T - \theta_{max}\right)^2}{2\sigma_e^2}\right) w\left(t; D\right)$$

The effective excitatory contrast for target plus background is slightly more complicated because there can be some contrast summation or cancellation depending on the phase and orientation of the target relative to the background

$$c_{TBe}\left(\mathbf{x}, t\right) = \lambda\left(\mathbf{x}\right)\sqrt{c_{Be}^2\left(\mathbf{x}, t\right) + c_{Te}^2\left(\mathbf{x}, t\right)}$$

where $\lambda\left(\mathbf{x}\right)$ is the contrast correction factor,

$$\lambda\left(\mathbf{x}\right) = \frac{C_{TB}\left(\mathbf{x}\right)}{\sqrt{C_B^2\left(\mathbf{x}\right) + C_T^2\left(\mathbf{x}\right)}}$$

The spatiotemporal impulse response function is the separable product of a Gaussian distribution and a gamma distribution

$$h_e\left(\mathbf{x}, t\right) = gauss\left(\mathbf{x}; s_e\right) gamma\left(t; a_e, b_e\right)$$

where $s_e$ is the standard deviation of the 2D Gaussian and $a_e, b_e$ are the two parameters of a gamma distribution: $gamma\left(t; a_e, b_e\right) = b_e^{-a_e} t^{a_e - 1} exp\left(-t/b_e\right)/\Gamma\left(a_e\right)$.

The formulas for the normalization signal are the same as for the excitatory signal, except the four parameters are allowed to differ: $\Omega_e = \left(\sigma_e, s_e, a_e, b_e\right)$, $\Omega_n = \left(\sigma_n, s_n, a_n, b_n\right)$.

Parameter values from known properties of single neurons in primary visual cortex were adopted. The response exponent $p$ was constrained to 2.0, consistent with single neuron contrast-response functions (**Albrecht and Hamilton, 1982**; **Geisler and Albrecht, 1997**; **Sclar et al., 1990**). It was assumed that the peak orientation of the excitatory signal and suppressive normalization signal $\theta_{max}$ were the same (**Cavanaugh et al., 2002a**), but that the orientation bandwidth of the normalization signal was greater than that of the excitation signal, $\sigma_n > \sigma_e$ (**Cavanaugh et al., 2002b**), and that the spatial pooling region for the normalization signal was larger than that for the excitation signal, $s_n > s_e$ (**Cavanaugh et al., 2002a**; **Cavanaugh et al., 2002b**; **Levitt and Lund, 2002**; **Sceniak et al., 2001**). As shown in **Figure 8—figure supplement 1**, we found that setting $s_n > s_e$ had little effect on the modeling of our empirical results (**Figure 5**); for simplicity, our final model assumed $s_n = s_e$. Finally, it was assumed that the temporal dynamics of the normalization signal, determined by parameters $a_n$ and $b_n$, are slower than those for the excitation signal, determined by parameters $a_e$ and $b_e$ (**Groen et al., 2022**; **Zhou et al., 2019**).

The following were the parameter values used in **Figure 8**: $r_0$=0.03125,

$$\Omega_e = \left(\sigma_e = 15°, a_e = 9, b_e = 8ms, s_e = 0.28°\right)$$
$$\Omega_n = \left(\sigma_n = 20°, a_n = 9, b_n = 10.64ms, s_n = 0.28°\right)$$

All analyses were done using Matlab R2018a.

## Statistics

Two animals were examined to verify the consistency of experimental approach and results. Multiple recordings were made from the same animals. The number of recordings were based on previous experience; no statistical method was used to predetermine sample size.

Statistical analyses were conducted in Matlab (R2018a).

## Acknowledgements

We thank We thank members of Seidemann and Geisler laboratories for their assistances with this project. This work was supported by NIH grants EY-016454 to ES, EY-024662 to WSG and ES, BRAIN U01-NS099720 to ES and WSG, and DARPA-NESD0-N66001-17-C-4012 to ES.

## Additional information

### Funding

| Funder | Grant reference number | Author |
|---|---|---|
| National Institutes of Health | EY-016454 | Eyal Seidemann |
| National Institutes of Health | EY-024662 | Eyal Seidemann Wilson S Geisler |
| BRAIN Initiative | U01-NS099720 | Eyal Seidemann Wilson S Geisler |
| Defense Advanced Research Projects Agency | DARPA-NESD0-N66001-17-C-4012 | Eyal Seidemann |

The funders had no role in study design, data collection and interpretation, or the decision to submit the work for publication.

### Author contributions

Spencer Chin-Yu Chen, Conceptualization, Data curation, Software, Formal analysis, Investigation, Visualization, Methodology, Writing - original draft, Writing - review and editing; Yuzhi Chen, Data curation, Software, Formal analysis, Investigation, Visualization, Methodology, Writing - review and editing; Wilson S Geisler, Conceptualization, Software, Formal analysis, Supervision, Funding acquisition, Investigation, Visualization, Methodology, Writing - original draft, Writing - review and editing; Eyal Seidemann, Conceptualization, Formal analysis, Supervision, Funding acquisition, Investigation, Visualization, Methodology, Writing - original draft, Writing - review and editing

### Author ORCIDs

Spencer Chin-Yu Chen (ID) http://orcid.org/0000-0003-0191-7315
Eyal Seidemann (ID) http://orcid.org/0000-0003-2841-5948

### Ethics

All procedures have been approved by the University of Texas Institutional Animal Care (IACUC protocol #AUP-2016-00274) and Use Committee and conform to NIH standards.

Reviewer #1 (Public Review): https://doi.org/10.7554/eLife.89570.3.sa1
Reviewer #2 (Public Review): https://doi.org/10.7554/eLife.89570.3.sa2
Author response https://doi.org/10.7554/eLife.89570.3.sa3

## Additional files

### Supplementary files

• Supplementary file 1. Supplementary perceptual masking demonstration.
• MDAR checklist

### Data availability

All data and script necessary to regenerate figures and statistics reported are available on Zenodo (https://doi.org/10.5281/zenodo.10815850).

The following dataset was generated:

| Author(s) | Year | Dataset title | Dataset URL | Database and Identifier |
|---|---|---|---|---|
| Chen SC, Chen Y, Geisler WS, Seidemann E | 2024 | Neural Correlates of Perceptual Similarity Masking in Primate V1 | https://doi.org/10.5281/zenodo.10815850 | Zenodo, 10.5281/zenodo.10815850 |

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
