## [Editor Report · eLife assessment]

This **important** study used Voltage Sensitive Dye Imaging (VSDI) to measure neural activity in the primary visual cortex of monkeys trained to detect an oriented grating target that was presented either alone or against an oriented mask. The authors show **convincingly** that the initial effect of the mask ran counter to the behavioral effects of the mask, a pattern that reversed in the latter phase of the response. They interpret these results in terms of influences from the receptive field center, and although an alternative view that emphasizes the role of the receptive field surround also seems reasonable, this study stands as an interesting contribution to our understanding of mechanisms of visual perception.

---

## [Referee Report · Reviewer #1 (Public Review)]

This is a clear account of some interesting work. The experiments and analyses seem well done and the data are useful. It is nice to see that VSDI results square well with those from prior extracellular recordings.

The authors have done a good job responding to the main points of my previous review. One important question remains, as stated in that review:

"My reading is that this is primarily a study of surround suppression with results that follow pretty directly from what we already know from that literature, and although they engage with some of the literature they do not directly mention surround suppression in the text. Their major effect - what they repeatedly describe as a "paradoxical" result in which the responses initially show a stronger response to matched targets and backgrounds and then reverse - seems to pretty clearly match the expected outcome of a stimulus that initially evokes additional excitation due to increased center contrast followed by slightly delayed surround suppression tuned to the same peak orientation. Their dynamics result seems entirely consistent with previous work, e.g. Henry at al 2020, particularly their Fig. 3 https://elifesciences.org/articles/54264, so it seems like a major oversight to not engage with that work at all, and to explain what exactly is new here."

Their rebuttal of my first review is not convincing -- I still believe that surround influences are important and perhaps predominant in determining the outcome of the experiments. This is particularly clear for the "paradoxical" dynamics that they observe, which seem exactly to reflect the behavior of the surround.

The authors' arguments to the contrary are based on three main points. First, their stimuli cover the center and surround, unlike those of many previous experiments, so they argue that this somehow diminishes the impact of the surround. But the argument is not accompanied by data showing the effects of center stimuli alone or surround stimuli alone. Second, their model -- a normalization model -- does not need surround influences to account for the masking effect. Third, they cite human psychophysical masking results from their collaborators (Sebastian et al 2017), but do not cite an equally convincing demonstration that surround contrast creates potent orientation selective masking when presented alone (Petrov et al 2005, https://doi.org/10.1523/JNEUROSCI.2871-05.2005).

At the end of the day, these issues will be resolved by further experiments, not argumentation. The paper stands as an excellent contribution, but it might be wise for the authors to be less doctrinaire in their interpretations.

---

## [Referee Report · Reviewer #2 (Public Review)]

Summary

In this experiment, Voltage Sensitive Dye Imaging (VSDI) was used to measure neural activity in macaque primary visual cortex in monkeys trained to detect an oriented grating target that was presented either alone or against an oriented mask. Monkeys' ability to detect the target (indicated by a saccade to its location) was impaired by the mask, with the greatest impairment observed when the mask was matched in orientation to the target, as is also the case in human observers. VSDI signals were examined to test the hypothesis that the target-evoked response would be maximally suppressed by the mask when it matched the orientation of the target. In each recording session, fixation trials were used to map out the spatial response profile and orientation domains that would then be used to decode the responses on detection trials. VSDI signals were analyzed at two different scales: a coarse scale of the retinotopic response to the target and a finer scale of orientation domains within the stimulus-evoked response. Responses were recorded in three conditions: target alone, mask alone, and target presented with mask. Analyses were focused on the target evoked response in the presence of the mask, defined to be the difference in response evoked by the mask with target (target present) versus the mask alone (target absent). These were computed across five 50 msec bins total, 250 msec, which was the duration of the mask (target present trials, 50% of trials) / mask + target (target present trials, 50% of trials). Analyses revealed that in an initial (transient) phase the target evoked response increased with similarity between target and mask orientation. As the authors note, this is surprising given that this was the condition where the mask maximally impaired detection of the target in behavior. Target evoked responses in a later ('sustained') phase fell off with orientation similarity, consistent with the behavioral effect. When analyzed at the coarser scale the target evoked response, integrated over the full 250 msec period showed a very modest dependence on mask orientation. The same pattern held when the data were analyzed on the finer orientation domain scale, with the effect of the mask in the transient phase running counter to the perceptual effect of the mask and the sustained response correlating the perceptual effect. The effect of the mask was more pronounced when analyzed at the scale.

Strengths

The work is on the whole very strong. The experiments are thoughtfully designed, the data collection methods are good, and the results are interesting. The separate analyses of data at a coarse scale that aggregates across orientation domains and a more local scale of orientation domains is a strength and it is reassuring that the effects at the more localized scale are more clearly related to behavior, as one would hope and expect. The results are strengthened by modeling work shown in Figure 8, which provides a sensible account of the population dynamics. The analyses of the relationship between VSDI data and behavior are well thought out and the apparent paradox of the anti-correlation between VSDI and behavior in the initial period of response, followed by a positive correlation in the sustained response period is intriguing.

---

## [Author Response]

The following is the authors’ response to the current reviews.

**eLife assessment**
This important study used Voltage Sensitive Dye Imaging (VSDI) to measure neural activity in the primary visual cortex of monkeys trained to detect an oriented grating target that was presented either alone or against an oriented mask. The authors show convincingly that the initial effect of the mask ran counter to the behavioral effects of the mask, a pattern that reversed in the latter phase of the response. They interpret these results in terms of influences from the receptive field center, and although an alternative view that emphasizes the role of the receptive field surround also seems reasonable, this study stands as an interesting and important contribution to our understanding of mechanisms of visual perception.
**Public Reviews:**

**Reviewer #1 (Public Review):**
This is a clear account of some interesting work. The experiments and analyses seem well done and the data are useful. It is nice to see that VSDI results square well with those from prior extracellular recordings.The authors have done a good job responding to the main points of my previous review. One important question remains, as stated in that review:"My reading is that this is primarily a study of surround suppression with results that follow pretty directly from what we already know from that literature, and although they engage with some of the literature they do not directly mention surround suppression in the text. Their major effect - what they repeatedly describe as a "paradoxical" result in which the responses initially show a stronger response to matched targets and backgrounds and then reverse - seems to pretty clearly match the expected outcome of a stimulus that initially evokes additional excitation due to increased center contrast followed by slightly delayed surround suppression tuned to the same peak orientation. Their dynamics result seems entirely consistent with previous work, e.g. Henry at al 2020, particularly their Fig. 3 https://elifesciences.org/articles/54264, so it seems like a major oversight to not engage with that work at all, and to explain what exactly is new here."Their rebuttal of my first review is not convincing -- I still believe that surround influences are important and perhaps predominant in determining the outcome of the experiments. This is particularly clear for the "paradoxical" dynamics that they observe, which seem exactly to reflect the behavior of the surround.The authors' arguments to the contrary are based on three main points. First, their stimuli cover the center and surround, unlike those of many previous experiments, so they argue that this somehow diminishes the impact of the surround. But the argument is not accompanied by data showing the effects of center stimuli alone or surround stimuli alone. Second, their model -- a normalization model -- does not need surround influences to account for the masking effect. Third, they cite human psychophysical masking results from their collaborators (Sebastian et al 2017), but do not cite an equally convincing demonstration that surround contrast creates potent orientation selective masking when presented alone (Petrov et al 2005, https://doi.org/10.1523/JNEUROSCI.2871-05.2005).At the end of the day, these issues will be resolved by further experiments, not argumentation. The paper stands as an excellent contribution, but it might be wise for the authors to be less doctrinaire in their interpretations.

We thank the reviewer for their positive comments and constructive criticism. In general, we agree with the reviewer’s comments. Importantly, we do not claim that there is no effect from the surround. What we say in the discussion is:

“Because our targets are added to the background rather than occluding it, it is likely that a significant portion of the behavioral and neural masking effects that we observe come from target-mask interactions at the target location rather than from the effect of the mask in the surround.”

We still stand by this assessment. We also make the point that, at least within the framework of our delayed normalization model, there is no need for the normalization mechanism to extend beyond the center mechanism to account for our results, and even if the normalization mechanism is somewhat larger than the center, the overlap region at the center would still have a large contribution to the modulations. Overall, we agree that these issues will be need to be resolved by future experiments.

For the reasons discussed in our previous reply, we disagree with the reviewers’ statement “…this is primarily a study of surround suppression with results that follow pretty directly from what we already know from that literature”. For similar reasons we disagree with the statement “It is nice to see that VSDI results square well with those from prior extracellular recordings”.

**Reviewer #2 (Public Review):**
SummaryIn this experiment, Voltage Sensitive Dye Imaging (VSDI) was used to measure neural activity in macaque primary visual cortex in monkeys trained to detect an oriented grating target that was presented either alone or against an oriented mask. Monkeys' ability to detect the target (indicated by a saccade to its location) was impaired by the mask, with the greatest impairment observed when the mask was matched in orientation to the target, as is also the case in human observers. VSDI signals were examined to test the hypothesis that the target-evoked response would be maximally suppressed by the mask when it matched the orientation of the target. In each recording session, fixation trials were used to map out the spatial response profile and orientation domains that would then be used to decode the responses on detection trials. VSDI signals were analyzed at two different scales: a coarse scale of the retinotopic response to the target and a finer scale of orientation domains within the stimulus-evoked response. Responses were recorded in three conditions: target alone, mask alone, and target presented with mask. Analyses were focused on the target evoked response in the presence of the mask, defined to be the difference in response evoked by the mask with target (target present) versus the mask alone (target absent). These were computed across five 50 msec bins total, 250 msec, which was the duration of the mask (target present trials, 50% of trials) / mask + target (target present trials, 50% of trials). Analyses revealed that in an initial (transient) phase the target evoked response increased with similarity between target and mask orientation. As the authors note, this is surprising given that this was the condition where the mask maximally impaired detection of the target in behavior. Target evoked responses in a later ('sustained') phase fell off with orientation similarity, consistent with the behavioral effect. When analyzed at the coarser scale the target evoked response, integrated over the full 250 msec period showed a very modest dependence on mask orientation. The same pattern held when the data were analyzed on the finer orientation domain scale, with the effect of the mask in the transient phase running counter to the perceptual effect of the mask and the sustained response correlating the perceptual effect. The effect of the mask was more pronounced when analyzed at the scale.StrengthsThe work is on the whole very strong. The experiments are thoughtfully designed, the data collection methods are good, and the results are interesting. The separate analyses of data at a coarse scale that aggregates across orientation domains and a more local scale of orientation domains is a strength and it is reassuring that the effects at the more localized scale are more clearly related to behavior, as one would hope and expect. The results are strengthened by modeling work shown in Figure 8, which provides a sensible account of the population dynamics. The analyses of the relationship between VSDI data and behavior are well thought out and the apparent paradox of the anti-correlation between VSDI and behavior in the initial period of response, followed by a positive correlation in the sustained response period is intriguing.

We thank the reviewer for their positive comments.

**Recommendations for the authors:**

**Reviewer #1 (Recommendations For The Authors):**
None, except perhaps for a more balanced representation of the "surround" possibility in the Discussion. The Petrov et al paper (https://doi.org/10.1523/JNEUROSCI.2871-05.2005) should be considered and cited.

As discussed above, we believe that our discussion of possible contribution from the surround is balanced. While the paper by Petrov et al is interesting, the stimuli used to study the surround effects are quite different (e.g., gap between center and surround, and the sharp edge of the surround inner boundary) so direct comparison with our results is not possible.

**Reviewer #2 (Recommendations For The Authors):**
The authors have addressed the questions/suggestions I raised in my review.

The following is the authors’ response to the original reviews.

We thank the reviewers for their helpful comments and suggestions.

**eLife assessment**
This is an important contribution that extends earlier single-unit work on orientation-specific center-surround interactions to the domain of population responses measured with Voltage Sensitive Dye (VSD) imaging and the first to relate these interactions to orientation-specific perceptual effects of masking. The authors provide convincing evidence of a pattern of results in which the initial effect of the mask seems to run counter to the behavioral effects of the mask, a pattern that reversed in the latter phase of the response. It seems likely that the physiological effects of masking reported here can be attributed to previously described signals from the receptive field surround.

We thank the reviewers for bringing up the relation of our results to findings from previous orientation-specific center-surround interactions studies. In our final manuscript, we added a paragraph discussing this important issue. Briefly, for multiple reasons, we believe that the orientation-dependent behavioral and neural masking effects that we observe are unlikely to depend on previously described center-surround interactions in V1. First, in human subjects, perceptual similarity masking effects are almost entirely accounted for by target-mask interactions at the target location and are recapitulated when the mask has the same size and location as the target (Sebastian et al 2017). Second, in our computational model, the effect of mask orientation on the dynamics of the response are qualitatively the same if the mask is restricted to the size and location of the target while mask contrast is increased (Fig. 8 – figure supplement 3). Third, in our model, the results are qualitatively the same when the spatial pooling region for the normalization signal is the same as that for the excitation signal (Fig. 8 – figure supplement figure 1). These considerations suggest that center-surround interactions may not be necessary for neural and behavioral similarity masking effects with additive targets.

We would also like to point out some key differences between the stimuli that we use and the ones used in most previous center-surround studies. First, in our experiments, the target and the mask were additive, while in most previous center-surround studies the target occludes the background. Such studies therefore restrict the mask effect to the surround, while in our study we allow target-mask interactions at the center. Second, most center-surround studies have a sharp-edged target/surround, while in our experiments no sharp edges were present. Unpublished results from our lab suggest that such sharp edges have a large impact on V1 population responses. A third key difference is that our stimuli were flashed for a short interval of 250 ms corresponding to a typical duration of a fixation in natural vision, while most previous center-surround studies used either longer-duration drifting stimuli or very short-duration random-order stimuli for reverse-correlation analysis.

In addition, we would like to emphasize that our results go beyond previous studies in two important ways. First, we study the effect of similarity masking in behaving animals and quantitatively compare the effect of similarity masking on behavior and physiology in the same subjects and at the same time. Second, VSD imaging allows us to capture the dynamics of superficial V1 population responses over the entire population of millions of neurons activated by the target at two important spatial scales. Such results therefore complement electrophysiological studies that examine the activity of a very small subset of the active neurons.

**Public Reviews:**

**Reviewer #1 (Public Review):**
This is a clear account of some interesting work. The experiments and analyses seem well done and the data are useful. It is nice to see that VSDI results square well with those from prior extracellular recordings. But the work may be less original than the authors propose, and their overall framing strikes me as odd. Some additional clarifications could make the contribution more clear.

Please see our reply above regarding the agreement with previous studies and framing.

My reading is that this is primarily a study of surround suppression with results that follow pretty directly from what we already know from that literature, and although they engage with some of the literature they do not directly mention surround suppression in the text. Their major effect - what they repeatedly describe as a "paradoxical" result in which the responses initially show a stronger response to matched targets and backgrounds and then reverse - seems to pretty clearly match the expected outcome of a stimulus that initially evokes additional excitation due to increased center contrast followed by slightly delayed surround suppression tuned to the same peak orientation. Their dynamics result seems entirely consistent with previous work, e.g. Henry et al 2020, particularly their Fig. 3 https://elifesciences.org/articles/54264, so it seems like a major oversight to not engage with that work at all, and to explain what exactly is new here.

We thank the reviewer for the pointing out this previous work which we now cite in the final version of the manuscript. For the reasons discussed above, while this study is interesting and related to our work, we believe that our results are quite distinct.

In the discussion (lines 315-316), they state "in order to account for the reduced neural sensitivity with target-background similarity in the second phase of the response, the divisive normalization signal has to be orientation selective." I wonder whether they observed this in their modeling. That is, how robust were the normalization model results to the values of sigma_e and sigma_n? It would be useful to know how critical their various model parameters were for replicating the experimental effects, rather than just showing that a good account is possible.

Thank you for this suggestion. In the final manuscript we include a supplementary figure that shows how the model’s predictions are affected by the orientation tuning and spatial extent of the normalization signal, and by the size and contrast of the mask (Fig. 8 – figure supplement 1-4).

The majority of their target/background contrast conditions were collected only in one animal. This is a minor limitation for work of this kind, but it might be an issue for some.

We agree that this is a limitation of the current study. These are challenging experiments and we were unable to collect all target/background contrast combinations from both monkeys. However, in the common conditions, the results appear similar in the two animals, and the key results seem to be robust to the contrast combination in the animal in which a wider range of contrast combinations was tested. We added these points to the discussion in the final manuscript.

The authors point out (line 193-195) that "Because the first phase of the response is shorter than the second phase, when V1 response is integrated over both phases, the overall response is positively correlated with the behavioral masking effect." I wonder if this could be explored a bit more at the behavioral level - i.e. does the "similarity masking" they are trying to explain show sensitivity to presentation time?

We agree that testing the effect of stimulus duration on similarity masking is interesting, but unfortunately, it is beyond the scope of the current study. We would also like to point out that the duration of the presentation was selected to match the typical time of fixation during natural behaviors, so much shorter or much longer stimulus durations would be less relevant for natural vision.

From Fig. 3 it looks like the imaging ROI may include some opercular V2. If so, it's plausible that something about the retinotopic or columnar windowing they used in analysis may remove V2 signals, but they don't comment. Maybe they could tell us how they ensured they only included V1?

We thank the reviewer for this comment. As part of our experiments, we extract a detailed retinotopic map for each chamber, so we were able to ensure that the area used for the decoding analysis lays entirely within V1. We now incorporate this information in the final manuscript (Fig. 3 – figure supplement 1).

In the discussion (lines 278-283) they say "The positive correlation between the neural and behavioral masking effects occurred earlier and was more robust at the columnar scale than at the retinotopic scale, suggesting that behavioral performance in our task is dominated by columnar scale signals in the second phase of the response. To the best of our knowledge, this is the first demonstration of such decoupling between V1 responses at the retinotopic and columnar scales, and the first demonstration that columnar scale signals are a better predictor of behavioral performance in a detection task." I am having trouble finding where exactly they demonstrate this in the results. Is this just by comparison of Figs. 4E,K and 5E,K? I may just be missing something here, but the argument needs to be made more clearly since much of their claim to originality rests on it.

We thank the reviewer for this comment. In the final manuscript we are more explicit when we discuss this point and refer to the relevant panels in Figs. 4, 5 and their figure supplements. To substantiate this key claim, we also report the timing of the transition between the two phases in all temporal correlation panels and report the neural-behavioral correlation for the integration period.

**Reviewer #2 (Public Review):**
SummaryIn this experiment, Voltage Sensitive Dye Imaging (VSDI) was used to measure neural activity in macaque primary visual cortex in monkeys trained to detect an oriented grating target that was presented either alone or against an oriented mask. Monkeys' ability to detect the target (indicated by a saccade to its location) was impaired by the mask, with the greatest impairment observed when the mask was matched in orientation to the target, as is also the case in human observers. VSDI signals were examined to test the hypothesis that the target-evoked response would be maximally suppressed by the mask when it matched the orientation of the target. In each recording session, fixation trials were used to map out the spatial response profile and orientation domains that would then be used to decode the responses on detection trials. VSDI signals were analyzed at two different scales: a coarse scale of the retinotopic response to the target and a finer scale of orientation domains within the stimulus-evoked response. Responses were recorded in three conditions: target alone, mask alone, and target presented with mask. Analyses were focused on the target evoked response in the presence of the mask, defined to be the difference in response evoked by the mask with target (target present) versus the mask alone (target absent). These were computed across five 50 msec bins total, 250 msec, which was the duration of the mask (target present trials, 50% of trials) / mask + target (target present trials, 50% of trials). Analyses revealed that in an initial (transient) phase the target evoked response increased with similarity between target and mask orientation. As the authors note, this is surprising given that this was the condition where the mask maximally impaired detection of the target in behavior. Target evoked responses in a later ('sustained') phase fell off with orientation similarity, consistent with the behavioral effect. When analyzed at the coarser scale the target evoked response, integrated over the full 250 msec period showed a very modest dependence on mask orientation. The same pattern held when the data were analyzed on the finer orientation domain scale, with the effect of the mask in the transient phase running counter to the perceptual effect of the mask and the sustained response correlating the perceptual effect. The effect of the mask was more pronounced when analyzed at the scale.StrengthsThe work is on the whole very strong. The experiments are thoughtfully designed, the data collection methods are good, and the results are interesting. The separate analyses of data at a coarse scale that aggregates across orientation domains and a more local scale of orientation domains is a strength and it is reassuring that the effects at the more localized scale are more clearly related to behavior, as one would hope and expect. The results are strengthened by modeling work shown in Figure 8, which provides a sensible account of the population dynamics. The analyses of the relationship between VSDI data and behavior are well thought out and the apparent paradox of the anti-correlation between VSDI and behavior in the initial period of response, followed by a positive correlation in the sustained response period is intriguing.Points to Consider / Possible ImprovementsThe biphasic nature of the relationship between neural and behavioral modulation by the mask and the surprising finding that the two are anticorrelated in the initial phase are left as a mystery. The paper would be more impactful if this mystery could be resolved.

We thank the reviewer for the positive comments. In our view, while our results are surprising, there may not be a remaining mystery that needs to be resolved. As our model shows, the biphasic nature of V1’s response can be explained by a delayed orientation-tuned gain control. Our results are consistent with the hypothesis that perception is based on columnar-scale V1 signals that are integrated over an approximately 200 ms long period that incorporates both the early and the late phase of the response, since such decoded V1 signals are positively correlated with the behavioral similarity masking effect (Fig. 5D, J; Fig. 5 – figure supplement 1). We now explain this more clearly in the discussion of our final manuscript.

The finding is based on analyses of the correlation between behavior and neural responses. This appears in the main body of the manuscript and is detailed in Figures S1 and S2, which show the correlation over time between behavior and target response for the retinotopic and columnar scale.One possible way of thinking of this transition from anti- to positive correlation with behavior is that it might reflect the dynamics of a competitive interaction between mask and target, with the initial phase reflecting predominantly the mask response, with the target emerging, on some trials, in the latter phase. On trials when the mask response is stronger, the probability of the target emerging in the latter phase, and triggering a hit, might be lower, potentially explaining the anticorrelation in the initial phase. The sustained response may be a mixture of trials on which the target response is or is not strong enough to overcome the effect of the mask sufficiently to trigger target detection.It would, I think, be worth examining this by testing whether target dynamics may vary, depending on whether the monkey detected the target (hit trials) or failed to detect the target (miss trials). Unless I missed it I do not think this analysis was done. Consistent with this possibility, the authors do note (lines 226-229) that "The trajectories in the target plus mask conditions are more complex. For example, when mask orientation is at +/- 45 deg to the target, the population response is initially dominated by the mask, but then in mid-flight, the population response changes direction and turns toward the direction of the target orientation." This suggests (to this reviewer, at least) that the emergence of a positive correlation between behavioral and neural effects in the latter phase of the response could reflect either a perceptual decision that the target is present or perhaps deployment of attention to the location of the target.It may be that this transition reflected detection, in which it might be more likely on hit trials than miss trials. Given the SNR it would presumably be difficult to do this analysis on a trial-by-trial basis, but the hit and miss trials (which make each make up about 1/2 of all trials) could be averaged separately to see if the mid-flight transition is more prominent on hit trials. If this is so for the +/- 45 degree case it would be good to see the same analysis for other combinations of target and mask. It would also be interesting to separate correct reject trials from false alarms, to determine whether the mid-flight transition tends to occur on false alarm trials.If these analyses do not reveal the predicted pattern, they might still merit a supplemental figure, for the sake of completeness.

We thank the reviewer for suggesting this interesting possibility. The original analysis in the manuscript was based on both correct and incorrect trials, raising the possibility that our results reflect some contribution from decision- and/or attention-related signals rather than from low-level nonlinear encoding mechanisms in V1 that we postulate in our model (Fig. 8). To explore this possibility, we re-examined our results while excluding error trials. We found that our key results from Figs 4 and 5 – namely that there is an early transient phase in which the neural and behavioral similarity effects are anti-correlated, and a later sustained phase in which they are positively correlated – hold even for the subset of correct trials, reducing the possibility that decision/attention-related signals play a major role in explaning our results. We now include the results of this analysis as a supplementary figure in the final manuscript (Fig. 4 – figure supplement 2). While there may be some interesting differences in the response dynamics between correct and incorrect trials, the current study was not designed to address this question and the large number of conditions and small number of repeats that it necessitated make this data set suboptimal for examining these phenomena.

References

Sebastian S, Abrams J, Geisler WS. 2017. Constrained sampling experiments reveal principles of detection in natural scenes. Proc Natl Acad Sci U S A 114: E5731-e40